# Sequential Monte Carlo for Policy Optimization in Continuous POMDPs

**Hany Abdulsamad**[*1,2] **Sahel Iqbal**[*2] **Simo Särkkä**[2]
[1]University of Amsterdam [2]Aalto University
{sahel.iqbal,simo.sarkka}@aalto.fi
h.abdulsamad@uva.nl

## Abstract

Optimal decision-making under partial observability requires agents to balance reducing uncertainty (exploration) against pursuing immediate objectives (exploitation). In this paper, we introduce a novel policy optimization framework for continuous partially observable Markov decision processes (POMDPs) that explicitly addresses this challenge. Our method casts policy learning as probabilistic inference in a non-Markovian Feynman–Kac model that inherently captures the value of information gathering by anticipating future observations, without requiring suboptimal approximations or handcrafted heuristics. To optimize policies under this model, we develop a nested sequential Monte Carlo (SMC) algorithm that efficiently estimates a history-dependent policy gradient under samples from the optimal trajectory distribution induced by the POMDP. We demonstrate the effectiveness of our algorithm across standard continuous POMDP benchmarks, where existing methods struggle to act under uncertainty.

## 1 Introduction

Optimal decision-making under uncertainty is central to building autonomous agents capable of operating in real-world environments. The ability to quantify and act on uncertainty enables agents to exhibit greater autonomy and robustness, allowing them to react effectively beyond controlled laboratory environments — a hallmark of adaptive systems [Feldbaum, 1963]. In practice, sensory limitations or environmental factors often conceal information required for optimal decision-making. This setting has been a long-standing challenge in the field of decision theory and the canonical framework to deal with such scenarios is that of partially observable Markov decision processes (POMDPs) [Åström, 1965, Aoki, 1967, Sondik, 1971].

The primary challenge of POMDPs lies in the entanglement of probabilistic inference and sequential planning. At each step, an agent must anticipate how its actions will shape future observations, what those observations may reveal about the latent state, and how this information can influence future decisions. Effective decision-making in POMDPs hence necessitates an exhaustive consideration of all possible state-action-observation trajectories over long horizons. While exact solutions exist for discrete POMDPs with small spaces [Cassandra et al., 1997, Porta et al., 2005, Poupart et al., 2006], continuous POMDPs pose far greater challenges, as the belief state becomes an infinite-dimensional object. Tractable solutions are restricted to linear-Gaussian problems with quadratic rewards [Aoki, 1967, Stengel, 1994], a special setting where inference and control decouple without loss of optimality [Bar-Shalom and Tse, 1974]. In more general continuous settings, existing methods often rely on problematic simplifications, such as an invalid decoupling of inference from control [Tse and Bar-Shalom, 1975, Li and Todorov, 2007], assuming maximum-likelihood future observations [Platt et al., 2010], or applying local linear-Gaussian approximations [Van Den Berg et al., 2012, Indelman et al., 2015].

---

[*]Equal contribution.

39th Conference on Neural Information Processing Systems (NeurIPS 2025).

Simulation-based methods have recently advanced learning in POMDPs by considering only a random subset of possible paths in the space of states, observations, and actions, thereby mitigating the *curse of histories* [Pineau et al., 2003]. This includes both trajectory-based planners [Hafner et al., 2019b, Wang et al., 2020] and deep reinforcement learning (RL) algorithms [Wierstra et al., 2007, Igl et al., 2018, Hafner et al., 2019a, Han et al., 2019, Lee et al., 2020, Meng et al., 2021, Zhang et al., 2023]. However, a weakness shared by many of these RL approaches is their adoption of the so-called QMDP approximation [Littman et al., 1995] to optimize policies based on latent-state dynamics, rather than reasoning over the space of beliefs. In QMDP, the latent state is assumed to be fully observable after a single step, thereby ignoring the impact of future observations on decision-making. As a result, *information gathering becomes incidental, rather than intentional*.

We propose a novel policy optimization algorithm for learning in continuous POMDPs. By directly reasoning over belief trajectories that account for future observations, our approach results in deliberate information gathering. It builds on the well-established connection between optimization and probabilistic inference [Dayan and Hinton, 1997, Attias, 2003, Kappen, 2005, Toussaint and Storkey, 2006, Todorov, 2008, Rawlik et al., 2012, Levine, 2018, Watson et al., 2020]. This framework is particularly well-suited for POMDPs, as it casts the coupled interaction between belief propagation and decision-making as a unified statistical inference problem. This perspective enables the use of state-of-the-art inference techniques, avoiding the common approximations that often limit traditional approaches. Our key contributions are:

- A Feynman–Kac model that captures the adaptive nature of decision-making in POMDPs and naturally incorporates the value of future observations.
- A nested sequential Monte Carlo (SMC) method that simulates adaptive decision-making by sampling from the optimal trajectory distribution of a POMDP.
- A policy optimization algorithm framed as maximum likelihood estimation within the Feynman–Kac model, resulting in a novel policy gradient method for POMDPs.

We begin by introducing the notation and objective of POMDPs. We then present our nested non-Markovian Feynman–Kac model, which captures the coupling between inference and decision-making in POMDPs. Building on this, we develop an efficient sampling scheme for policy optimization. Finally, we relate our approach to prior work and evaluate it on standard POMDP benchmarks.

## 2    Partially Observable Markov Decision Processes

A partially observable Markov decision process is defined by the tuple $(\mathcal{S}, \mathcal{A}, \mathcal{Z}, R, f, g)$. Here, $\mathcal{S} \subset R^{d_s}$, $\mathcal{A} \subset R^{d_a}$, and $\mathcal{Z} \subset R^{d_z}$ are the sets of allowed states, actions, and observations, respectively. At any time $t \in \mathbb{N}$, the transition model $f : \mathcal{S} \times \mathcal{A} \times \mathcal{S} \to \mathbb{R}_{\geq 0}$ defines the likelihood of moving to a state $s_{t+1}$ after taking action $a_t$ in state $s_t$, denoted as $f(s_{t+1} \mid s_t, a_t)$. The initial state follows $s_0 \sim p(s_0)$. The reward function $R_t : \mathcal{S} \times \mathcal{A} \to \mathbb{R}$ assigns a reward to a state-action pair $(s_t, a_{t-1})^2$, and the observation model $g : \mathcal{S} \times \mathcal{Z} \to \mathbb{R}_{\geq 0}$ is the probability of observing $z_t$ when the state is $s_t$, written as $g(z_t \mid s_t)$. In a POMDP, the true state $s_t$ is latent. The agent instead maintains a *belief state* $p(s_t \mid z_{0:t}, a_{0:t-1})$ given a history of observations $z_{0:t}$ and actions $a_{0:t-1}$, with $a_{0:-1} = \emptyset$ by convention. Finally, a policy is a mapping from observation-action histories to actions. We consider stochastic policies $\pi_\phi(a_t \mid z_{0:t}, a_{0:t-1})$ parameterized by $\phi$.

Modern approaches for POMDPs [Igl et al., 2018, Lee et al., 2020, Hafner et al., 2019a, Meng et al., 2021, Chen et al., 2022] commonly adopt a learning objective $\phi^* = \mathrm{argmax}_\phi \mathcal{J}(\phi)$, where $\mathcal{J}(\phi) = \mathbb{E}_{p_\phi} \left[ \sum_{t=1}^{T} R_t(s_t, a_{t-1}) \right]$ is the expected cumulative reward under the joint density $p_\phi(s_{0:T}, z_{0:T}, a_{0:T-1})$ of all random variables:

$$p_\phi(\cdot) := p(s_0) \left\{ \prod_{t=0}^{T-1} f(s_{t+1} \mid s_t, a_t)\, \pi_\phi(a_t \mid z_{0:t}, a_{0:t-1}) \right\} \left\{ \prod_{k=0}^{T} g(z_k \mid s_k) \right\}.$$

We refer to the objective $\mathcal{J}(\phi)$ as the *state-space objective*, as it is defined in terms of the state-action reward $R_t(\cdot)$ and the generative process $p_\phi(\cdot)$ induced by the state transition dynamics $f(\cdot)$. We

---

[2]For notational convenience, and without loss of generality, we consider a transition-based reward definition $R_t(s_t, a_{t-1}, s_{t-1})$ [Sutton and Barto, 2018], in which the dependency on $s_{t-1}$ is omitted.

denote this underlying process as the *state-space generative process*. While valid, the objective $\mathcal{J}(\phi)$ obscures the adaptive nature of the decision-making process by not explicitly conditioning on intermediate observations. This makes it difficult to reason about how the agent's beliefs evolve over time, an essential aspect in POMDPs [Kaelbling et al., 1998, Thrun et al., 2005]. In practice, optimal decision-making in POMDPs involves two intertwined components, as described by the dual control framework [Bar-Shalom and Tse, 1974]: a *backward-looking* inference process, which updates the belief state $p(s_t \mid z_{0:t}, a_{0:t-1})$ using realized observations, and a *forward-looking* planning process, which selects actions to maximize the expected future reward $\mathcal{J}_{t:T}(\phi)$ under the predictive distribution $p_\phi(s_{t:T}, a_{t:T-1}, z_{t+1:T} \mid z_{0:t}, a_{0:t-1})$. Crucially, accounting for future observations $z_{t+1:T}$ enables exploration, as it encourages the agent to consider how future, yet-to-be-seen information will affect its beliefs and, consequently, its actions.

To model adaptivity, we depart from paradigms that rely on state-space generative processes. Instead, we draw on the work of Thrun [1999] and construct an adaptive Monte Carlo simulation procedure which we denote as the *belief-space generative process*. We demonstrate that the state-space objective $\mathcal{J}(\phi)$ is equivalent to a belief-space objective $\mathcal{B}(\phi)$, which is governed by the belief-space generative process that explicitly tracks and updates the belief.

**Proposition 1** (Belief-Space Objective). *The expected cumulative reward objective $\mathcal{J}(\phi)$ defined over a sequence of state-action reward functions $R_t(s_t, a_{t-1})$ evaluated according to a state-space generative process $p_\phi(s_{0:T}, z_{0:T}, a_{0:T-1})$ is equivalent to*

$$\mathcal{B}(\phi) = \mathbb{E}_{p_\phi(z_{0:T}, a_{0:T-1})} \left[ \sum_{t=1}^{T} \ell_t(z_{0:t}, a_{0:t-1}) \right], \tag{1}$$

*where $\ell_t(z_{0:t}, a_{0:t-1}) := \mathbb{E}_{p(s_t \mid z_{0:t}, a_{0:t-1})} \left[ R_t(s_t, a_{t-1}) \right]$ is the expected reward under the belief $p(s_t \mid z_{0:t}, a_{0:t-1})$, and $p_\phi(z_{0:T}, a_{0:T-1})$ characterizes the belief-space generative process:*

$$p_\phi(z_{0:T}, a_{0:T-1}) = p(z_0) \prod_{t=0}^{T-1} p(z_{t+1} \mid z_{0:t}, a_{0:t}) \, \pi_\phi(a_t \mid z_{0:t}, a_{0:t-1}), \tag{2}$$

*with $p(z_0) = \mathbb{E}_{p(s_0)} \left[ g(z_0 \mid s_0) \right]$ and*

$$p(z_{t+1} \mid z_{0:t}, a_{0:t}) = \iint_{\mathcal{S}^2} g(z_{t+1} \mid s_{t+1}) \, f(s_{t+1} \mid s_t, a_t) \, p(s_t \mid z_{0:t}, a_{0:t-1}) \, \mathrm{d}s_t \, \mathrm{d}s_{t+1}.$$

The proof is in Appendix A.1. Proposition 1 defines an objective that factorizes according to the causal structure of decision making: an agent's belief $p(s_t \mid \cdot)$ depends only on past quantities. This reformulation expresses the sequential decision-making problem explicitly in terms of histories $(z_{0:t}, a_{0:t-1})$ and highlights how agents are incentivized to explore by steering the belief towards informative observations that improve the expected utility $\ell_t(\cdot)$ [Kaelbling et al., 1998]. Next, we leverage this belief-space generative process and objective within a probabilistic inference framework.

## 3 POMDPs as Feynman–Kac Models

We present a novel technique to simulate adaptive decision-making and learning in POMDPs. We leverage the connection between optimization and inference [Toussaint and Storkey, 2006, Levine, 2018] to construct a Feynman–Kac (FK) model that emulates the decision-making process in the belief space (1), and derive a policy gradient approach for policy optimization. We start with a brief introduction to Feynman–Kac models and sequential Monte Carlo.

### 3.1 Feynman–Kac Models and Sequential Monte Carlo

Let $x_{0:T}$ be a sequence of random variables taking values in $\mathcal{X}^{(T+1)}$. Their joint probability density $\mathbb{M}_T(x_{0:T})$ can be decomposed as $\mathbb{M}_T(x_{0:T}) = \mathbb{M}_0(x_0) \prod_{t=1}^{T} M_t(x_t \mid x_{0:t-1})$, where $\mathbb{M}_0$ is the prior law of $x_0$ and $M_t$ are transition kernels. Now, let $G_0 : \mathcal{X} \to \mathbb{R}^+$ and $G_t : \mathcal{X}^{(t+1)} \to \mathbb{R}^+$ for $t \in \{1, \ldots, T\}$ be so-called potential functions. Then the sequence of distributions

$$\mathbb{Q}_t(x_{0:t}) := \frac{1}{L_t} \mathbb{M}_0(x_0) \, G_0(x_0) \prod_{k=1}^{t} M_k(x_k \mid x_{0:k-1}) \, G_k(x_{0:k}), \quad t \in \{0, \ldots, T\}, \tag{3}$$

defines a *Feynman–Kac model* [Del Moral, 2004] on $\mathcal{X}^{(T+1)}$, with $L_t$ being the normalizing constant.

Sequential Monte Carlo (SMC) algorithms or *particle filters* provide discrete approximations to Feynman–Kac models [Chopin and Papaspiliopoulos, 2020]. Here we present the *bootstrap* particle filter from Gordon et al. [1993]. We start by drawing i.i.d. samples $x_0^n \sim \mathbb{M}_0$ with weights $w_0^n \propto G_0(x_0^n)$ for $n \in \{1, \ldots, N\}, N > 1$, and alternate between the following steps for $t \in \{1, \ldots T\}$:

1. *Resample:* Sample $A_t^n \sim \text{Categorical}(N, \{w_{t-1}^n\}_{n=1}^N)$.
2. *Mutate:* Generate importance samples $x_t^n \sim M_t(\cdot \,|\, x_{0:t-1}^{A_t^n})$.
3. *Reweight:* Assign importance weights $w_t^n = W_t^n / \sum_{n=1}^N W_t^n$, where $W_t^n = G_t(x_{0:t}^n)$.

Here, $x_{0:t}^n$ is defined recursively as $x_{0:t}^n := (x_{0:t-1}^{A_t^n}, x_t^n)$. The algorithm returns a discrete approximation $\sum_{n=1}^N w_t^n \delta(x_{0:t} - x_{0:t}^n) \approx \mathbb{Q}_t(x_{0:t})$ at each time step $t$, where $\delta$ is the Dirac delta function. It also provides an unbiased estimate of the marginal likelihood increment $L_t/L_{t-1} \approx \frac{1}{N} \sum_{n=1}^N W_t^n$. For $t = T$, samples from the particle filter approximate the terminal *smoothing distribution* $\mathbb{Q}_T(x_{0:T})$.

### 3.2  Probabilistic Inference for POMDPs

Proposition 1 defines a sequential decision-making problem over observation-action histories, resembling objectives from stochastic control and reinforcement learning [Aoki, 1967, Sutton and Barto, 2018]. This allows us to draw on established techniques for solving such problems. In particular, since inference and decision-making are inherently coupled in POMDPs, we adopt the control-as-inference perspective [Toussaint and Storkey, 2006, Levine, 2018] and construct a Feynman–Kac model over $(z_{0:t}, a_{0:t-1})$ trajectories. This probabilistic formulation yields a surrogate of the belief-space objective that serves as the foundation for our policy learning approach.

Let $\{\mathcal{O}_t\}_{t>0}$ be a sequence of auxiliary binary random variables with a history-dependent likelihood

$$p(\mathcal{O}_t = 1 \,|\, z_{0:t}, a_{0:t-1}) \propto \exp\left\{\eta\, \ell_t(z_{0:t}, a_{0:t-1})\right\}, \quad t \in \{1, \ldots, T\}, \tag{4}$$

where $\ell_t(\cdot)$ is the expected reward function specified in Proposition 1 and $\eta > 0$ is a constant. We assume the reward function $R_t$ (and hence $\ell_t$) is bounded. $\{\mathcal{O}_t = 1\}$ represents the event that the pair $(z_t, a_{t-1})$ is optimal given the partial trajectory $(z_{0:t-1}, a_{0:t-2})$, and a higher probability of optimality corresponds to higher expected reward $\ell_t$. The exponential function is strictly positive and monotonic. Positive potential functions are required by the Feynman–Kac formalism and monotonicity guarantees that the potential function maintains the same maximizer as the reward function. For brevity, we write $\mathcal{O}_t$ to denote the event $\{\mathcal{O}_t = 1\}$ and $\mathcal{O}_{1:t} := \cup_{k=1}^t \{\mathcal{O}_k = 1\}$ to denote optimality of an entire horizon. With the dynamic process (2) and potential functions (4), we construct an FK model

$$\begin{aligned}
\Psi_t(z_{0:t}, a_{0:t-1}; \phi) &:= p_\phi(z_{0:t}, a_{0:t-1} \,|\, \mathcal{O}_{1:t}) \\
&= \frac{1}{p_\phi(\mathcal{O}_{1:t})}\, p(z_0) \left\{ \prod_{k=0}^{t-1} p(z_{k+1} \,|\, z_{0:k}, a_{0:k}) \right\} \\
&\quad \times \left\{ \prod_{l=0}^{t-1} \pi_\phi(a_l \,|\, z_{0:l}, a_{0:l-1}) \right\} \left\{ \prod_{m=1}^t p(\mathcal{O}_m \,|\, z_{0:m}, a_{0:m-1}) \right\},
\end{aligned} \tag{5}$$

for $t \in \{1, \ldots, T\}$. $\Psi_t$ represents the joint distribution over observation-action trajectories *conditioned on optimality* until time $t$. The normalizing constant

$$p_\phi(\mathcal{O}_{1:t}) = \int p(\mathcal{O}_{1:t} \,|\, z_{0:t}, a_{0:t-1})\, p_\phi(z_{0:t}, a_{0:t-1})\, \mathrm{d}z_{0:t}\, \mathrm{d}a_{0:t-1}$$

quantifies the probability of generating optimal trajectories under the policy $\pi_\phi$, making it a natural optimization target in the control-as-inference framework [Toussaint and Storkey, 2006].

This alternative objective serves as a surrogate for the POMDP objective $\mathcal{B}(\phi)$ and can be justified from multiple perspectives. Optimizing the evidence lower bound [ELBO, Blei et al., 2017] of this objective is equivalent to solving a maximum entropy RL problem [Ziebart et al., 2008, Rawlik et al., 2012, Levine, 2018]. Alternatively, directly maximizing $\log p_\phi(\mathcal{O}_{1:t})$ corresponds to optimizing an

**Algorithm 1:** Particle POMDP Policy Optimization (P3O).

---

**input:** Initial policy parameters $\phi_0$, step size sequence $\{\alpha_k\}_{k \in \mathbb{N}}$, and number of samples $N$.
**output:** Optimal policy parameters $\phi^*$.

**1** Set $k \leftarrow 0$.
**2** **while** *not converged* **do**
**3**     Sample $\{(z_{0:T}^i, a_{0:T-1}^i)\}_{i=1}^N$ approximately distributed as $\Psi_T(\cdot; \phi_k)$.    // Algorithm 2
**4**     Estimate the policy gradient $\hat{g}_k \leftarrow \frac{1}{N} \sum_{i=1}^N \sum_{t=0}^{T-1} \nabla_\phi \log \pi_\phi(a_t^i \mid z_{0:t}^i, a_{0:t-1}^i)|_{\phi=\phi_k}$.
**5**     Update parameters $\phi_{k+1} \leftarrow \phi_k + \alpha_k \hat{g}_k$ and set $k \leftarrow k + 1$.

---

optimistic risk-sensitive objective [Marcus et al., 1997, Rawlik, 2013, Watson et al., 2020]:

$$\mathcal{B}_\eta(\phi) := \frac{1}{\eta} \log p_\phi(\mathcal{O}_{1:t}) = \frac{1}{\eta} \log \mathbb{E}_{p_\phi(z_{0:T}, a_{0:T-1})} \left[ \exp \left\{ \eta \sum_{t=1}^T \ell_t(z_{0:t}, a_{0:t-1}) \right\} \right]. \quad (6)$$

The hyperparameter $\eta$ controls the degree of risk, with the risk-neutral objective $\mathcal{B}(\phi)$ (1) recovered in the limit $\eta \to 0$. In this work, we maximize the objective $\mathcal{B}_\eta(\phi)$ directly using a Monte Carlo policy gradient algorithm, which we describe next.

### 3.3 Particle POMDP Policy Optimization (P3O)

The risk-sensitive belief-space objective $\mathcal{B}_\eta(\phi)$ is proportional to the log-normalizing constant $\log p_\phi(\mathcal{O}_{1:T})$. As noted in Section 3.1, particle filters provide unbiased estimates of the normalizing constant, hinting at a natural approach to policy learning: obtain gradients by differentiating through a particle filter and use gradient ascent. However, the gradient cannot be obtained this way because standard resampling schemes are discontinuous. While differentiable resampling schemes exist [Corenflos et al., 2021], they do not easily extend to non-Markovian systems. We circumvent this issue by estimating the policy gradient directly using Fisher's identity [Cappé et al., 2005].

**Proposition 2** (POMDP Policy Gradient). *Consider the risk-sensitive belief-space objective $\mathcal{B}_\eta(\phi)$ from (6) and the distribution $\Psi_T(z_{0:T}, a_{0:T-1}; \phi)$ from (5). Under suitable regularity conditions, the pathwise policy gradient of $\mathcal{B}_\eta(\phi)$ satisfies*

$$\nabla_\phi \mathcal{B}_\eta(\phi) \propto \mathbb{E}_{\Psi_T(z_{0:T}, a_{0:T-1}; \phi)} \left[ \sum_{t=0}^{T-1} \nabla_\phi \log \pi_\phi(a_t \mid z_{0:t}, a_{0:t-1}) \right]. \quad (7)$$

The proof is available in Appendix A.2. Proposition 2 provides a principled way to estimate the gradient of the objective $\mathcal{B}_\eta(\phi)$ with respect to the policy parameters $\phi$ by integrating $\nabla_\phi \log \pi_\phi$ under the Feynman–Kac distribution $\Psi_T$. This integral is intractable, so we propose constructing a particle filter targeting $\Psi_T$ and using the resulting Monte Carlo samples to estimate the policy gradient [Kantas et al., 2015]. The meta procedure is outlined in Algorithm 1.

While a Monte Carlo estimate of (7) resembles the REINFORCE estimator [Williams, 1992], it differs significantly. REINFORCE is the gradient of the *risk-neutral* objective $\mathcal{B}(\phi)$ given by:

$$\nabla_\phi \mathcal{B}(\phi) \propto \mathbb{E}_{p_\phi(z_{0:T}, a_{0:T-1})} \left[ \sum_{t=0}^{T-1} \left( \sum_{k=1}^T \ell_k(z_{0:k}, a_{0:k-1}) \right) \nabla_\phi \log \pi_\phi(a_t \mid z_{0:t}, a_{0:t-1}) \right],$$

where $p_\phi(z_{0:T}, a_{0:T-1})$ is the *prior* policy-induced joint distribution. Monte Carlo estimates of this expectation typically suffer from high variance, as the prior distribution $p_\phi(\cdot)$ is often uninformative with respect to the reward and rarely samples from high-reward regions. In contrast, our method samples trajectories directly from the *reward-weighted posterior* distribution $\Psi_T(\cdot; \phi)$, which concentrates probability mass on high-reward trajectories due to conditioning on the optimality events $\mathcal{O}_{1:T}$ (4). As a result, our estimator benefits from lower variance relative to REINFORCE. When using sequential Monte Carlo approximations of $\Psi_T(z_{0:T}, a_{0:T-1}; \phi)$, which we describe in the upcoming section, this variance reduction arises naturally from an importance (re)-sampling correction, yielding a more efficient estimator for the policy gradient.

**Algorithm 2:** Particle filter to sample from $\Psi_T(z_{0:T}, a_{0:T-1}; \phi)$.

---

**input:** Number of history particles $N$, number of belief particles $M$, and temperature $\eta$.
**output:** Weighted particle set $\{z_{0:T}^n, a_{0:T-1}^n, v_T^n\}_{n=1}^N$ approximating $\Psi_T(z_{0:T}, a_{0:T-1}; \phi)$.
**notation:** Operations indexed by $n$ and $m$ are run over $n = 1, \ldots, N$ and $m = 1, \ldots, M$.

1   Sample belief particles $s_0^{nm} \sim p_0(s_0)$ and set $w_0^{nm} \leftarrow 1/M$.            // Initialize

2   Sample observation $z_0^n \sim \sum_{m=1}^M w_0^{nm} g(z_0 \mid s_0^{nm})$ and set $v_0^n \leftarrow 1/N$.

3 **for** $t \leftarrow 0$ **to** $T-1$ **do**

4      Sample history ancestor indices: $A_t^n \sim \text{Categorical}(N, \{v_t^n\}_{n=1}^N)$.      // Resample

5      Sample belief ancestor indices: $B_t^{nm} \sim \text{Categorical}(M, \{w_t^{nm}\}_{m=1}^M)$.

6      Replace history particles: $\{z_{0:t}^n, a_{0:t-1}^n, v_t^n\} \leftarrow \{z_{0:t}^{A_t^n}, a_{0:t-1}^{A_t^n}, 1/N\}$.

7      Replace belief particles: $\{s_t^{nm}, w_t^{nm}\} \leftarrow \{s_t^{A_t^n B_t^{nm}}, 1/M\}$.

8      Sample action: $a_t^n \sim \pi_\phi(\cdot \mid z_{0:t}^n, a_{0:t-1}^n)$.                 // Mutate

9      Propagate belief particles: $s_{t+1}^{nm} \sim f(\cdot \mid s_t^{nm}, a_t^n)$.

10     Sample observation: $z_{t+1}^n \sim \sum_{m=1}^M w_t^{nm} g(\cdot \mid s_{t+1}^{nm})$.

11     Weight and normalize belief particles: $w_{t+1}^{nm} \propto w_t^{nm} g(z_{t+1}^n \mid s_{t+1}^{nm})$.      // Reweight

12     Estimate expected reward: $\ell_{t+1}(z_{0:t+1}^n, a_{0:t}^n) \approx \sum_{m=1}^M w_{t+1}^{nm} R_{t+1}(s_{t+1}^{nm}, a_t^n)$.

13     Weight and normalize history particles: $v_{t+1}^n \leftarrow v_t^n \exp\left\{\eta\, \ell_{t+1}(z_{0:t+1}^n, a_{0:t}^n)\right\}$.

---

Implementing a particle filter for $\Psi_T$ requires sampling from the sequence of target distributions $\Psi_t$ defined by the Feynman–Kac model in (5). A critical challenge in POMDPs is that the construction of these target distributions depends on the intermediate belief distributions $p(s_t \mid z_{0:t}, a_{0:t-1})$, as both the utility function $\ell_t(z_{0:t}, a_{0:t-1})$ and the predictive process $p(z_{t+1} \mid z_{0:t}, a_{0:t})$ are implicitly functions of the belief. To address this dependency, we propose nesting two particle filters. The outer filter targets the trajectory-level distribution defined by the Feynman–Kac model, while the inner filter maintains an estimate of the belief states required at each step. We use the outer samples to optimize the parameters $\phi$ via the policy gradient method in Algorithm 1. We describe the nested particle filter in detail in the next section.

**Remark 1.** *The target distribution $\Psi_T$ (5) admits the following decomposition:*

$$\Psi_T(z_{0:T}, a_{0:T-1}) \propto p(z_0 \mid \mathcal{O}_{1:T}) \prod_{t=0}^{T-1} p(z_{t+1} \mid z_{0:t}, a_{0:t}, \mathcal{O}_{t+1:T}) \frac{p(\mathcal{O}_{t+1:T} \mid z_{0:t}, a_{0:t})}{p(\mathcal{O}_{t+1:T} \mid z_{0:t}, a_{0:t-1})}. \quad (8)$$

*The terms $\log p(\mathcal{O}_{t+1:T} \mid [z_{0:t}, a_{0:t-1}])$ and $\log p(\mathcal{O}_{t+1:T} \mid [z_{0:t}, a_{0:t-1}], a_t)$ stand for the soft value functions $V_t([z_{0:t}, a_{0:t-1}])$ and $Q_t([z_{0:t}, a_{0:t-1}], a_t)$ in the control-as-inference framework, as they describe probabilities of future optimality given partial trajectories [Levine, 2018]. Soft actor-critic methods learn parametrized approximations of these functions [Haarnoja et al., 2018, Lee et al., 2020, Zhang et al., 2023]. In contrast, we sample from $\Psi_T$ directly using a particle filter, bypassing the need to learn explicit value functions. This decomposition is detailed in Appendix A.3.*

### 3.4   A Particle Filter for the Distribution $\Psi_T$

Following the policy optimization algorithm described in the previous section, the final component required by our method is a procedure to generate samples from the target trajectory distribution $\Psi_T(z_{0:T}, a_{0:T-1}; \phi)$. This section introduces a novel nested particle filtering scheme that enables such sampling efficiently, thereby supporting the implementation of Algorithm 1.

As illustrated in Section 3.1, a particle filter approximates Feynman–Kac models defined by a sequence of transition kernels $M_t$ and potential functions $G_t$. Comparing our target distribution in (5) to the model in (3), we observe that $\Psi_t$ corresponds to a Feynman–Kac model with $M_t = p_\phi(z_t, a_{t-1} \mid z_{0:t-1}, a_{0:t-2})$ and $G_t = \exp\left\{\eta\, \ell_t(z_{0:t}, a_{0:t-1})\right\}$. However, direct application of particle filtering is not feasible because here both $M_t$ and $G_t$ require computing expectations with respect to the belief distribution $p(s_t \mid z_{0:t}, a_{0:t-1})$, which is generally not available in closed form.

To address this, and inspired by the success of *nested* sequential Monte Carlo algorithms in parameter estimation problems [Chopin et al., 2013, Crisan and Míguez, 2018], we propose nesting two (bootstrap) particle filters with *interleaved* updates. The first particle filter, denoted as the *belief filter*, maintains a set of $M$ weighted particles $\{s_t^m, w_t^m\}_{m=1}^M$ approximating the posterior $p(s_t \,|\, z_{0:t}, a_{0:t-1}) \approx \sum_{m=1}^M w_t^m \delta(s_t - s_t^m)$ according to weights updated recursively using the observation model, $w_t^m \propto w_{t-1}^m g(z_t \,|\, s_t^m)$. The second filter, denoted as the *Feynman–Kac filter*, targets $\Psi_T$, while maintaining a set of $N$ weighted particles $\{z_{0:t}^n, a_{0:t-1}^n, v_t^n, \{s_t^{nm}, w_t^{nm}\}_{m=1}^M\}_{n=1}^N$ with the associated weights updated according to the expected reward $v_t^n \propto v_{t-1}^n \ell_t(z_{0:t}^n, a_{0:t-1}^n)$. Notice that the state of the Feynman–Kac filter contains the full states of $N$ unique and independent belief filters, each associated with a history particle $(z_{0:t}^n, a_{0:t-1}^n)$. These belief filters are used within the Feynman–Kac filter to approximate the reward $\ell_t(z_{0:t}, a_{0:t-1}) \approx \sum_{m=1}^M w_t^{nm} R_t(s_t^{nm}, a_{t-1}^n)$ and predictive observation distribution $p(z_{t+1} \,|\, z_{0:t}, a_{0:t}) \approx \sum_{m=1}^M w_t^{nm} g(z_{t+1} \,|\, s_{t+1}^{nm})$ according to their definition in Proposition 1. Algorithm 2 outlines the detailed filter operations.

This interleaved construction of particle filters yields a set $\left\{z_{0:T}^n, a_{0:T-1}^n, v_T^n\right\}_{n=1}^N$ of weighted samples that are approximately distributed according to $\Psi_T$, and can be used to estimate the policy gradient in Algorithm 1. Nonetheless, such samples may suffer from a problem known as path degeneracy: as $T$ increases, all trajectories tend to coalesce and share a common ancestor at the initial time step [Del Moral and Miclo, 2001]. This can lead to higher variance of the policy gradient (7). To mitigate this, an additional backward sampling step [Godsill et al., 2004] can be applied after performing Algorithm 2, yielding more diverse trajectories. We detail this procedure in Appendix C.

Algorithm 2 has computational complexity $\mathcal{O}(NMT)$, which increases to $\mathcal{O}(NM^2T^2)$ when employing backward sampling (Appendix C). For long-horizon problems, we therefore recommend avoiding backward sampling and instead tuning the parameter $\eta$ to mitigate particle degeneracy.

# 4 Related Work

A substantial body of literature addresses *online* planning in POMDPs. These algorithms plan from the current belief, execute the chosen action, incorporate the new observation, and then repeat the cycle [Silver and Veness, 2010, Ross et al., 2008, Somani et al., 2013, Kurniawati and Yadav, 2016, Ye et al., 2017]. Although this loop makes them flexible, it also imposes a significant computational cost, since the planner is invoked after every observation. This procedure may violate the real-time requirements of certain systems. Moreover, most online planners are designed for discrete POMDPs and rely on Monte Carlo Tree Search (MCTS) variants [Kocsis and Szepesvári, 2006, Browne et al., 2012]. While these MCTS-based solvers have been successfully extended to continuous domains [Sunberg and Kochenderfer, 2018, Lim et al., 2021], such approaches rely on heuristics, like progressive widening, to artificially limit the branching factor of MCTS.

In contrast to online planners, a separate class of *offline* algorithms learn value functions and corresponding policies to select optimal actions without replanning [Kaelbling et al., 1998, Thrun, 1999]. Estimating value functions in continuous POMDPs is difficult because they are defined over infinite-dimensional belief spaces. Therefore, Littman et al. [1995] introduced the *QMDP approximation*, which replaces *belief-space* Q-values with the expectation of fully observable *state-space* Q-values under the current belief, see Appendix B.1 and B.2 for more details. This simplification ignores the value of future information, preventing the agent from performing directed exploration, and is therefore sub-optimal [Ross et al., 2008]. Despite this, QMDP underpins many popular modern deep-RL algorithms [Hafner et al., 2019a, Zhang et al., 2019, Wang et al., 2020, Lee et al., 2020, Chen et al., 2022, Singh et al., 2021, Zhang et al., 2023]. A handful of approaches, such as those proposed by Igl et al. [2018], Meng et al. [2021] and Yang and Nguyen [2021], avoid the QMDP approximation, yet they still deviate from Bellman's optimality principles for POMDPs. Their learning pipelines interact directly with the environment and estimate the belief state and the returns via Monte Carlo rollouts driven by the state dynamics, rather than the belief dynamics. We discuss this difference in further detail in Appendix B.1 and B.3.

Our approach shares technical aspects with prior work. Particle filters have been used in several POMDP solvers [Thrun, 1999, Coquelin et al., 2008, Igl et al., 2018, Wang et al., 2020, Ma et al., 2020, Deglurkar et al., 2023], primarily to approximate the belief state. Our method differs in its use of a nested particle filtering procedure with interleaved updates that not only tracks the evolving

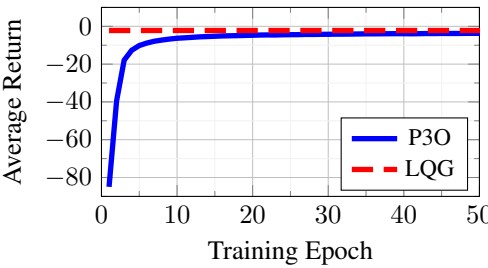 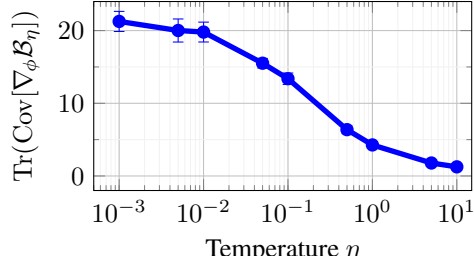

Figure 1: Benchmarking P3O on a linear-Gaussian quadratic problem. Left: LQG provides a tractable optimal baseline and P3O converges to a similar performance under general assumptions. Right: the influence of the temperature $\eta$ on the variance of the gradient, higher $\eta$ leads to smaller variance.

belief but also samples optimal observation–action trajectories for policy learning. While Lee et al. [2020], Wang et al. [2020], Zhang et al. [2023] also adopt a control-as-inference perspective, they rely on the sub-optimal QMDP approximation, which undermines directed exploration.

## 5  Numerical Evaluation

For empirical validation, we compare our method, *particle POMDP policy optimization* (P3O), against several popular baselines. *Stochastic latent actor-critic* [SLAC, Lee et al., 2020] combines the QMDP approximation with the soft actor-critic [SAC, Haarnoja et al., 2018] algorithm to learn a state-action value function and a history-dependent policy. *Deep variational reinforcement learning for POMDPs* [DVRL, Igl et al., 2018] avoids QMDP and learns a belief-dependent policy and value function. We replace its original advantage actor-critic [A2C, Mnih et al., 2016] training procedure with SAC to ensure parity. Finally, *Dual sequential Monte Carlo* [DualSMC, Wang et al., 2020] employs the QMDP approximation to learn a state-action value function and a belief-dependent policy via SAC, but also includes a planning step that selects actions based on their estimated advantage. To isolate algorithmic differences in policy learning, we equip all four methods with the same particle filter belief tracker, which has oracle access to the true observation models. Complete experimental details are given in Appendix D and implementations of all algorithms are available at https://github.com/Sahel13/particle-pomdp.

Our P3O-framework is agnostic to the policy representation: the policy can be explicitly history-dependent, $\pi_\phi\left(a_t \mid z_{0:t}, a_{0:t-1}\right)$, or belief-dependent, $\pi_\phi\left(a_t \mid b_t\right)$ where $b_t := p(s_t \mid z_{0:t}, a_{0:t-1})$. History-dependent policies are theoretically appealing, yet training long-horizon recurrent networks often suffers from vanishing gradients [Pascanu et al., 2013]. Belief-dependent policies sidestep this issue by acting on the instantaneous belief, which is a *sufficient statistic* of the entire observation–action history in POMDPs [Sondik, 1971, Kaelbling et al., 1998]. However, finite-sample approximations of belief have serious drawbacks [Thrun, 1999]: the same belief state can be approximated by different sets of particles, and neural networks usually cannot respect this invariance. Although this mismatch can introduce spurious effects, particle-based belief inputs have been empirically effective [Igl et al., 2018]; consequently, we explore both variants of the policy.

**Tractable LQG:** We validate our approach against a closed-form linear-quadratic Gaussian (LQG) benchmark. In this setting, belief propagation is computed exactly using a Kalman filter, and the optimal policy is an affine function of the Gaussian belief mean [Aoki, 1967]. We test P3O on a two-dimensional system with linear-Gaussian transition and observation models and a quadratic reward, where LQG provides a reference for the achievable cumulative reward. Figure 1 compares this baseline with P3O, which operates without exploiting any analytic structure. As training progresses, P3O converges toward the optimal performance of LQG. We also use this problem to illustrate the role of the temperature parameter $\eta$ in controlling the variance of the policy gradient in Figure 1. Larger values of $\eta$ reduce variance but introduce bias through the risk-sensitive objective, while $\eta \to 0$ recovers the unbiased REINFORCE gradient at the cost of high variance.

**Control Tasks:** Here we evaluate on two classical control tasks: stochastic, partially-observed variants of the pendulum and cart-pole swing-up tasks. In both settings, the agent cannot observe the velocity components of the state. These tasks are representative of standard control problems, where active information gathering is not critical. Hence, it is expected that the QMDP approximation should not be detrimental to learning. This intuition is confirmed by the results in Figure 2. The differences

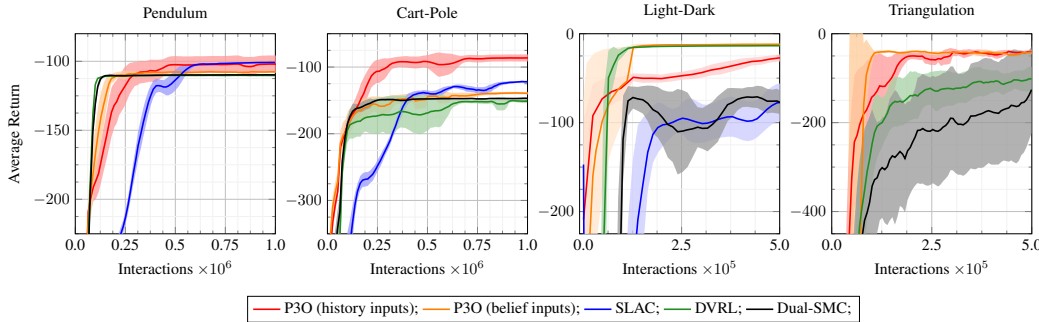

Figure 2: Experiment results on various benchmarks. We report the average return using 1024 trajectory rollouts and plot the mean and standard error over 10 training seeds. P3O and DVRL consistently outperform the QMDP-based solvers, SLAC and DualSMC, which especially struggle in the light-dark and triangulation tasks that require deliberate exploration.

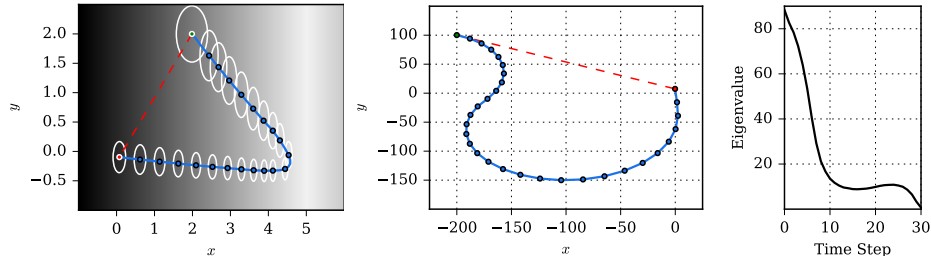

Figure 3: Example trajectories from the light-dark (left) and triangulation (middle and right) tasks. In the light-dark environment, the agent learns to steer towards low-uncertainty regions to localize itself, then moves to the target. We visualize the agent's belief at every time step. In the triangulation task, the agent executes a specialized maneuver to estimate its position using only heading measurements. We plot the largest eigenvalue of the agent's belief covariance over time. The red dashed lines represent the shortest path the agent could follow if no information gathering was required.

in convergence speed reflect how frequently gradient steps are performed. P3O and SLAC, which use recurrent policies over full histories, require complete trajectory rollouts before each update, unlike DVRL, DualSMC, and P3O with a belief-dependent policy. While belief-dependent policies can learn faster, their performance can be constrained by the quality of the belief representation, and (bootstrap) particle filters are known to produce degenerate approximations in certain cases [Bickel et al., 2008]. This trade-off is evident in the more challenging cart-pole environment.

**Light-Dark:** Next, we consider a continuous light-dark navigation task [Platt et al., 2010, Van Den Berg et al., 2012], where an agent must reach a goal state in a two-dimensional plane while relying on observations with location-dependent noise. Outside a *light* region, the observation noise is very large, whereas near the light the agent can localize accurately. The optimal strategy requires first navigating towards the light to reduce uncertainty before committing to the goal. QMDP-based baselines, DualSMC and SLAC, are doubly handicapped in this task. Their disregard for information-gathering results in persistently high-variance observations. This in turn leads to high-variance bootstrap targets in their Q-learning procedures, inflating gradient noise and stalling learning, as Figure 2 shows. In contrast, methods that explicitly reason over the belief space, such as P3O, learn to gather information, then exploit, achieving better performance. Figure 3 depicts an example trajectory generated by a policy learned with P3O. The policy intentionally steers away from the target to reduce uncertainty before moving towards the goal state.

**Triangulation:** In our final experiment, we consider an active triangulation task, in which the agent must reach the origin in a two-dimensional plane relying solely on heading measurements [Tse and Bar-Shalom, 1975]. This task is particularly difficult, as accurate distance estimation to the target requires the agent to perform a special maneuver to collect heading measurements from varied positions in the state space. As with the previous experiment, this environment has significant noise both in transitions and observations. P3O successfully learns to solve this task, see Figure 2. A representative trajectory from a policy learned with P3O is shown in Figure 3, exhibiting a characteristic S-shape oscillation around the line of sight.

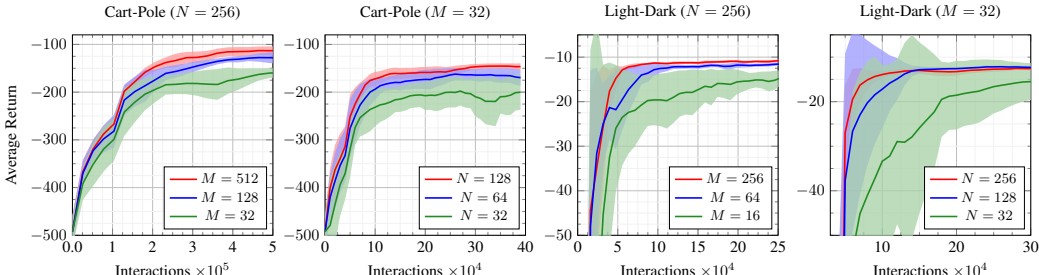

Figure 4: Influence of the number of particles in the Feynman–Kac filter $(N)$ and belief filter $(M)$. Increasing $N$ and $M$ improves the approximation of the target distribution $\Psi_T$ and the policy gradient, resulting in better learning performance. Plots depict the mean and standard error over 10 seeds.

**Ablation Studies:** Finally, we conduct ablation studies to disentangle the effect of P3O's hyperparameters. In Figure 4, we vary the number of particles used in the Feynman–Kac and belief filters. Increasing the particle count yields more accurate approximations of the target distribution $\Psi_T$ and accelerates learning, though with diminishing returns. In Figure 5, we examine the impact of the temperature parameter $\eta$. Proper tuning of $\eta$ is essential: it controls gradient variance and determines the effectiveness of policy updates, although too high values of $\eta$ can exacerbate path degeneracy in SMC.

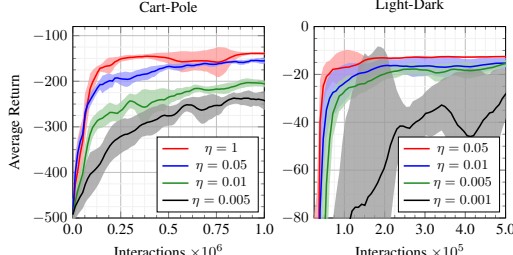

Figure 5: The effect of the temperature $\eta$ on policy learning. Lower values of $\eta$ lead to high-variance policy gradients and slower convergence.

## 6   Discussion

Optimal decision-making in partially observable environments presents fundamental challenges, as agents have to reason in the space of histories and beliefs. While many existing methods simplify this complexity using approximations such as QMDP, they typically sacrifice the ability to actively gather information and perform directed exploration. In this work, we introduced particle POMDP policy optimization (P3O), a principled policy learning algorithm that operates directly on the belief space. By formulating the POMDP learning problem within a Feynman–Kac framework, our approach integrates belief tracking with reward-guided trajectory sampling, enabling principled and efficient policy optimization under partial observability in continuous domains.

Our experiments, while focused on relatively low-dimensional tasks, highlight a fundamental limitation of several sophisticated algorithms that fail to learn in the light-dark and triangulation tasks. These tasks expose a key distinction between *dual-effect* systems, where actions influence both state transitions and the agent's uncertainty, and *neutral systems*, where actions do not affect state estimation [Feldbaum, 1963, Bar-Shalom and Tse, 1974]. In dual-effect settings, P3O demonstrates a clear advantage by explicitly incorporating exploration. In contrast, in neutral systems, QMDP-based algorithms like SLAC and DualSMC are likely to scale better.

Our method is not without limitations. First, the bootstrap particle filter used in our implementation may not scale well in high-dimensional settings due to increased discrepancy between the proposal and target distributions. Using more sophisticated proposals that take the likelihood into account [Doucet et al., 2000] or learning better proposals [Guarniero et al., 2017, Naesseth et al., 2018] may be necessary to ensure satisfactory performance for high-dimensional problems. Second, the particle belief-tracker (though not the policy learning loop) requires access to the oracle of the observation likelihood. Incorporating a learning-based state estimator that can operate from raw interaction alone is an important direction for future work. Third, quantifying the bias in the score estimate in Algorithm 2 is a challenging problem that we do not address. While standard SMC methods have well-understood bias properties ($\mathcal{O}(1/N)$ with $N$ particles for additive functionals [Del Moral, 2004]), our nested structure creates more complex bias behavior. Lastly, tuning the risk parameter $\eta$ is an open problem. We refer the reader to Watson and Peters [2022] for potential calibration strategies.

# 7 Contribution Statement

The original idea and motivation are due to HA. The methodology, proofs, and experiments were developed jointly by HA and SI, with equal contribution. SI and HA drafted the manuscript. SS provided regular feedback through weekly meetings, contributed valuable insights on the setting and background literature, and proofread the manuscript.

# 8 Acknowledgments

Part of this work was conducted while HA was a postdoctoral researcher at Aalto University and the Finnish Center for Artificial Intelligence (FCAI). HA also acknowledges funding from the Amsterdam Machine Learning Lab (AMLAB) at the University of Amsterdam. SI gratefully acknowledges funding from the Research Council of Finland. The authors thank Adrien Corenflos for proofreading the manuscript and providing valuable comments.

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

# A   Proofs and Derivations

## A.1   Proof of Proposition 1

*Proof.* We begin with the original definition of the objective function $\mathcal{J}(\phi)$:

$$\mathcal{J}(\phi) = \mathbb{E}_{p_\phi(z_{0:T}, a_{0:T-1}, s_{0:T})} \left[ \sum_{t=1}^{T} R_t(s_t, a_{t-1}) \right],$$

which defines the expected cumulative reward over a fixed horizon, where the expectation is over full trajectories $(z_{0:T}, a_{0:T-1}, s_{0:T})$ given a policy $\pi_\phi$. By linearity of expectation, we can interchange the expectation and the sum,

$$\mathcal{J}(\phi) = \sum_{t=1}^{T} \mathbb{E}_{p_\phi(z_{0:T}, a_{0:T-1}, s_{0:T})} \left[ R_t(s_t, a_{t-1}) \right].$$

Now, consider the term $\mathbb{E}_{p_\phi(\cdot)} \left[ R_t(s_t, a_{t-1}) \right]$. The reward $R_t$ depends only on the state $s_t$ and the previous action $a_{t-1}$. We can thus marginalize out variables corresponding to future times $t' > t$ and past states $s_{0:t-1}$:

$$\mathcal{J}(\phi) = \sum_{t=1}^{T} \mathbb{E}_{p_\phi(z_{0:t}, a_{0:t-1}, s_t)} \left[ R_t(s_t, a_{t-1}) \right]$$

$$= \sum_{t=1}^{T} \mathbb{E}_{p(s_t \mid z_{0:t}, a_{0:t-1}) \, p_\phi(z_{0:t}, a_{0:t-1})} \left[ R_t(s_t, a_{t-1}) \right],$$

where $p(s_t \mid z_{0:t}, a_{0:t-1})$ is the belief state and $p_\phi(z_{0:t}, a_{0:t-1})$ is the marginal probability of the history. Now, we can apply the law of total expectation to decouple the expectation over the history $(z_{0:t}, a_{0:t-1})$ from the expectation over the state $s_t$ leading to:

$$\mathcal{B}(\phi) := \mathbb{E}_{p_\phi(z_{0:T}, a_{0:T-1})} \left[ \sum_{t=1}^{T} \mathbb{E}_{p(s_t \mid z_{0:t}, a_{0:t-1})} \left[ R_t(s_t, a_{t-1}) \right] \right].$$

The outer expectation with respect to $p_\phi(z_{0:T}, a_{0:T-1})$ averages over all possible histories of observations and actions generated under the policy $\pi_\phi$. The inner expectation with respect to $p(s_t \mid \cdot)$ averages the reward $R_t$ over the possible states $s_t$, given a specific history $(z_{0:t}, a_{0:t-1})$. This objective is consistent with the definition of POMDP objectives according to Kaelbling et al. [1998]. To simplify the expression, we define an auxiliary function $\ell_t$ representing the expected immediate reward at time $t$ given the history up to that point:

$$\ell_t(z_{0:t}, a_{0:t-1}) := \mathbb{E}_{p(s_t \mid z_{0:t}, a_{0:t-1})} \left[ R_t(s_t, a_{t-1}) \right].$$

This definition effectively integrates out the dependency on the latent state $s_t$ by averaging according to the belief state distribution. Substituting this definition back yields

$$\mathcal{B}(\phi) = \mathbb{E}_{p_\phi(z_{0:T}, a_{0:T-1})} \left[ \sum_{t=1}^{T} \ell_t(z_{0:t}, a_{0:t-1}) \right],$$

with $p_\phi(z_{0:T}, a_{0:T-1})$ being the belief-space generative process

$$p_\phi(z_{0:T}, a_{0:T-1}) = p(z_0) \prod_{t=0}^{T-1} p(z_{t+1} \mid z_{0:t}, a_{0:t}) \, \pi_\phi(a_t \mid z_{0:t}, a_{0:t-1}),$$

where $p(z_0) = \mathbb{E}_{p(s_0)} \left[ g(z_0 \mid s_0) \right]$ and

$$p(z_{t+1} \mid z_{0:t}, a_{0:t}) = \iint_{\mathcal{S}^2} g(z_{t+1} \mid s_{t+1}) \, f(s_{t+1} \mid s_t, a_t) \, p(s_t \mid z_{0:t}, a_{0:t-1}) \, \mathrm{d}s_t \, \mathrm{d}s_{t+1},$$

as required. □

## A.2 Proof of Proposition 2

*Proof.* We begin with the definition of the risk-sensitive belief-space objective

$$\mathcal{B}_\eta(\phi) := \frac{1}{\eta} \log p_\phi(\mathcal{O}_{1:T}).$$

Since $\eta > 0$ is just a scalar, we focus on the gradient of the log-marginal likelihood $\log p_\phi(\mathcal{O}_{1:T})$

$$\begin{aligned}
\nabla_\phi \mathcal{B}_\eta(\phi) &\propto \nabla_\phi \log p_\phi(\mathcal{O}_{1:T}) \\
&= \frac{1}{p_\phi(\mathcal{O}_{1:T})} \nabla_\phi p_\phi(\mathcal{O}_{1:T}) \\
&= \frac{1}{p_\phi(\mathcal{O}_{1:T})} \nabla_\phi \int p_\phi(z_{0:T}, a_{0:T-1})\, p(\mathcal{O}_{1:T} \mid z_{0:T}, a_{0:T-1})\, \mathrm{d}z_{0:T}\, \mathrm{d}a_{0:T-1}.
\end{aligned}$$

Under suitable regularity conditions, we can interchange differentiation and integration (see Mohamed et al. [2020] for details on when this is admissible)

$$\nabla_\phi \mathcal{B}_\eta(\phi) \propto \frac{1}{p_\phi(\mathcal{O}_{1:T})} \int \nabla_\phi\, p_\phi(z_{0:T}, a_{0:T-1})\, p(\mathcal{O}_{1:T} \mid z_{0:T}, a_{0:T-1})\, \mathrm{d}z_{0:T}\, \mathrm{d}a_{0:T-1}.$$

Next, using the log-ratio trick, $\nabla f = f \nabla \log f$, the expression becomes:

$$\nabla_\phi \mathcal{B}_\eta(\phi) \propto \frac{1}{p_\phi(\mathcal{O}_{1:T})} \int p_\phi(z_{0:T}, a_{0:T-1}, \mathcal{O}_{1:T}) \nabla_\phi \log p_\phi(z_{0:T}, a_{0:T-1})\, \mathrm{d}z_{0:T}\, \mathrm{d}a_{0:T-1}$$

Now, let's recall the definition of $p_\phi(z_{0:T}, a_{0:T-1})$ from (2)

$$p_\phi(z_{0:T}, a_{0:T-1}) = p(z_0) \prod_{t=0}^{T-1} p(z_{t+1} \mid z_{0:t}, a_{0:t})\, \pi_\phi(a_t \mid z_{0:t}, a_{0:t-1}),$$

and the definition of $\Psi_T$ from (5)

$$\Psi_T(z_{0:T}, a_{0:T-1}; \phi) := p_\phi(z_{0:T}, a_{0:T-1} \mid \mathcal{O}_{1:T}) = \frac{p_\phi(z_{0:T}, a_{0:T-1}, \mathcal{O}_{1:T})}{p_\phi(\mathcal{O}_{1:T})}.$$

Plugging them back into the gradient expression, we get:

$$\begin{aligned}
\nabla_\phi \mathcal{B}_\eta(\phi) &\propto \mathbb{E}_{\Psi_T} \left[ \nabla_\phi \log \left\{ p(z_0) \prod_{t=0}^{T-1} p(z_{t+1} \mid z_{0:t}, a_{0:t})\, \pi_\phi(a_t \mid z_{0:t}, a_{0:t-1}) \right\} \right] \\
&= \mathbb{E}_{\Psi_T} \left[ \nabla_\phi \log \left\{ \prod_{t=0}^{T-1} \pi_\phi(a_t \mid z_{0:t}, a_{0:t-1}) \right\} + \nabla_\phi \log \left\{ p(z_0) \prod_{t=0}^{T-1} p(z_{t+1} \mid z_{0:t}, a_{0:t}) \right\} \right] \\
&= \mathbb{E}_{\Psi_T} \left[ \sum_{t=0}^{T-1} \nabla_\phi \log \pi_\phi(a_t \mid z_{0:t}, a_{0:t-1}) \right],
\end{aligned}$$

as required. $\qquad\square$

## A.3 Derivation of Equation (8)

We omit the dependence on $\phi$, and start from the definition of $\Psi_T$ in (5),

$$\begin{aligned}
\Psi_T(z_{0:T}, a_{0:T-1}) &= p(z_{0:T}, a_{0:T-1} \mid \mathcal{O}_{1:T}) \\
&= p(z_0 \mid \mathcal{O}_{1:T}) \prod_{t=0}^{T-1} p(a_t \mid z_{0:t}, a_{0:t-1}, \mathcal{O}_{1:T})\, p(z_{t+1} \mid z_{0:t}, a_{0:t}, \mathcal{O}_{1:T}) \\
&= p(z_0 \mid \mathcal{O}_{1:T}) \prod_{t=0}^{T-1} p(z_{t+1} \mid z_{0:t}, a_{0:t}, \mathcal{O}_{t+1:T})\, p(a_t \mid z_{0:t}, a_{0:t-1}, \mathcal{O}_{t+1:T}).
\end{aligned}$$

For the final step, we used the fact that $z_{t+1}$ and $a_t$ are conditionally independent of $\mathcal{O}_{1:t}$ given $(z_{0:t}, a_{0:t-1})$. Using Bayes' rule, we have

$$p(a_t \mid z_{0:t}, a_{0:t-1}, \mathcal{O}_{t+1:T}) = \pi(a_t \mid z_{0:t}, a_{0:t-1}) \frac{p(\mathcal{O}_{t+1:T} \mid z_{0:t}, a_{0:t})}{p(\mathcal{O}_{t+1:T} \mid z_{0:t}, a_{0:t-1})},$$

where $\pi(a_t \mid z_{0:t}, a_{0:t-1})$ is the prior (unconditioned) policy. Putting everything together, we get

$$\Psi_T(z_{0:T}, a_{0:T-1}) \propto p(z_0 \mid \mathcal{O}_{1:T}) \prod_{t=0}^{T-1} p(z_{t+1} \mid z_{0:t}, a_{0:t}, \mathcal{O}_{t+1:T}) \frac{p(\mathcal{O}_{t+1:T} \mid z_{0:t}, a_{0:t})}{p(\mathcal{O}_{t+1:T} \mid z_{0:t}, a_{0:t-1})},$$

as required.

# B  Decision-Making in POMDPs

## B.1  Bellman's Optimality Equations

In this section, we briefly review the fundamentals of optimal decision-making in partially observable Markov decision processes. We use $[z_{0:t}, a_{0:t-1}]$ to denote the history up to time $t$ as a stacked vector. The solution to a POMDP can be formulated in terms of optimal value functions, $V_t$ and $Q_t$, over beliefs and actions $([z_{0:t}, a_{0:t-1}], a_t)$ as follows [Thrun et al., 2005, Chapter 15]:

$$V_t([z_{0:t}, a_{0:t-1}]) := \max_{a_t \in \mathcal{A}} \ Q_t([z_{0:t}, a_{0:t-1}], a_t), \tag{9}$$

$$Q_t([z_{0:t}, a_{0:t-1}], a_t) := \int p(z_{t+1} \mid z_{0:t}, a_{0:t}) \tag{10}$$
$$\times \Big[ \ell_{t+1}(z_{0:t+1}, a_{0:t}) + V_{t+1}([z_{0:t+1}, a_{0:t}]) \Big] \, \mathrm{d}z_{t+1}$$

where $\ell_{t+1}(z_{0:t+1}, a_{0:t}) = \mathbb{E}_{p(s_{t+1} \mid z_{0:t+1}, a_{0:t})} \big[ R_{t+1}(s_{t+1}, a_t) \big]$ is the expected reward defined in Proposition 1 and $p(s_{t+1} \mid z_{0:t+1}, a_{0:t})$ is the updated belief after following action $a_t$ and observing $z_{t+1}$, retrieved according to a Bayes filter [Särkkä and Svensson, 2023]:

$$p(s_{t+1} \mid z_{0:t+1}, a_{0:t}) \propto g(z_{t+1} \mid s_{t+1}) \int f(s_{t+1} \mid s_t, a_t) \, p(s_t \mid z_{0:t}, a_{0:t-1}) \, \mathrm{d}s_t.$$

Unlike in fully observable Markov decision processes (MDPs), where value functions are defined over finite-dimensional spaces, POMDP value functions $V_t$ and $Q_t$ are defined over the infinite-dimensional space of histories. This makes solving Bellman's optimality equations significantly more challenging. Crucially, the value of information gathering is implicitly encoded into the value function through the predictive distribution over future observations

$$p(z_{t+1} \mid z_{0:t}, a_{0:t}) := \iint g(z_{t+1} \mid s_{t+1}) \, f(s_{t+1} \mid s_t, a_t) \, p(s_t \mid z_{0:t}, a_{0:t-1}) \, \mathrm{d}s_t \, \mathrm{d}s_{t+1},$$

so that the expected utility of an action $a_t$ depends not only on immediate rewards but also its effect on future belief refinement. This way, an agent can account for the long-term value of observations that improve its understanding of the latent state and thus influence subsequent decisions.

## B.2  The QMDP Approximation

To simplify Bellman's optimality equations in POMDPs, Littman et al. [1995] introduced what is known as the QMDP approximation, which replaces (10) with the approximation

$$Q_t([z_{0:t}, a_{0:t-1}], a_t) \approx \mathbb{E}_{p(s_t \mid z_{0:t}, a_{0:t-1})} \Big[ Q_t(s_t, a_t) \Big]$$
$$= \iint p(s_t \mid z_{0:t}, a_{0:t-1}) \, f(s_{t+1} \mid s_t, a_t) \Big[ R_{t+1}(s_{t+1}, a_t) + V_{t+1}(s_{t+1}) \Big] \, \mathrm{d}s_{t+1} \, \mathrm{d}s_t,$$

where $Q_t(s_t, a_t)$ is the state-space Q-function of the corresponding fully observable MDP. While this approach is algorithmically simpler, it relies on a strong, crude assumption: action values are estimated as if the latent state will be perfectly observable from the next time step onwards. This

effectively ignores the need for, and potential value of, future information gathering, as it fails to account for how future observations $z_{t+1}$ might refine the agent's knowledge of the latent state and improve decision-making. Consequently, this method decouples the procedures of probabilistic inference and decision-making, which are inherently intertwined in POMDPs, making any reduction in belief uncertainty merely incidental, rather than a directed outcome of the policy. For this reason, methods that use QMDP fail to address the core POMDP challenge of actively balancing the need to reduce state uncertainty (exploration) against maximizing expected extrinsic rewards based on the current state of belief (exploitation). Since optimal POMDP policies must achieve this balance, policies derived via the QMDP approximation are inherently sub-optimal [Ross et al., 2008]. State-of-the-art work on POMDPs such as [Hafner et al., 2019b,a, Lee et al., 2020, Wang et al., 2020] either explicitly or implicitly assume the QMDP setting.

### B.3 The Belief-Space Generative Process

The Bellman equations in (9) and (10) are computationally tractable only for a narrow class of continuous POMDPs. Consequently, state-of-the-art methods typically rely on sampling-based approximations. A critical distinction arises between reinforcement learning in MDPs and POMDPs, namely in the generative process used to obtain Monte Carlo samples.

In an MDP, Bellman's optimality principle defines the value of a state, $V_{t+1}(s_{t+1})$, as the expected sum of future rewards. This expectation is taken over the known state-transition dynamics $f(s_{t+1} \mid s_t, a_t)$. Hence, MDP algorithms interact directly with the transition dynamics to generate the state-space Monte Carlo tuples of the form $(s_t, a_t, s_{t+1}, r_{t+1})$, where $r_{t+1} = R_{t+1}(s_{t+1}, a_t, s_t)$. These samples are then used to approximate the expectation underpinning the state-value.

In contrast, optimal decision-making in POMDPs must operate in the space of histories or beliefs. According to Bellman's principle in POMDPs (10), the value function, $V_{t+1}([z_{0:t+1}, a_{0:t}])$, is evaluated in expectation over the observation predictive distribution $p(z_{t+1} \mid z_{0:t}, a_{0:t})$. Additionally, the corresponding reward signal is the expected reward, $l_{t+1} = \ell_{t+1}(z_{0:t+1}, a_{0:t})$, given the agent's belief. This implies that a theoretically sound sampling-based approach must simulate from this predictive process and collect the tuples of the form $([z_{0:t}, a_{0:t-1}], a_t, z_{t+1}, l_{t+1})$ to arrive at valid Monte Carlo estimations of the corresponding value functions.

However, many state-of-the-art reinforcement learning methods for POMDPs appear to deviate from this simulation process [Igl et al., 2018, Meng et al., 2021, Yang and Nguyen, 2021] and instead optimize the *state-space* objective via a state-space sampling process as described in Section 2. Instead of simulating from $p(z_{t+1} \mid z_{0:t}, a_{0:t})$, these methods sample observations by interacting with the transition dynamics $f(\underline{s}_{t+1} \mid \underline{s}_t, a_t)$ and effectively drawing from the conditional $g(z_{t+1} \mid \underline{s}_{t+1})$, where $\underline{s}_{t+1}$ is the *true, latent* state. Furthermore, they use the raw reward signal, $\underline{r}_{t+1} = R_{t+1}(\underline{s}_{t+1}, a_t, \underline{s}_t)$ associated with that specific latent state transition, rather than the expected reward, $l_{t+1} = \ell_{t+1}(z_{0:t+1}, a_{0:t})$, associated with agent's belief. This leads to a learning procedure that evolves with state-space dynamics, rather than belief-space dynamics.

## C   Backward Sampling

### C.1   Overview

Backward sampling is an algorithm used to generate more diverse samples from the smoothing distribution [Godsill et al., 2004, Lindsten and Schön, 2013]. While the smoothing distribution can be represented by the ancestral lineage of the trajectories generated by the particle filter, backward sampling prescribes an additional step: compute the likelihood that each particle from the previous time step could have led to a given current particle, and sample an ancestor from this distribution. As a result, backward sampling can generate trajectories that were not explicitly formed during the forward filtering pass.

We use the notation for Feynman–Kac models from Section 3.1 and follow the presentation in Corenflos [2024, Chapter 3]. The formula for backward sampling is based on the trivial identity

$$\mathbb{Q}_T(x_{0:t} \mid x_{t+1:T}) \propto \mathbb{Q}_T(x_{0:T}) = \frac{\mathbb{Q}_T(x_{0:T})}{\mathbb{Q}_t(x_{0:t})} \mathbb{Q}_t(x_{0:t}).$$

After performing particle filtering, we will have weighted particles $\{x_{0:t}^n, w_t^n\}_{n=1}^N$ that form a discrete approximation to the filtering distribution $\mathbb{Q}_t(x_{0:t})$ for all $t \in \{0, \ldots, T\}$. We can use these to form an approximation to $\mathbb{Q}_T(x_{0:t} \mid x_{t+1:T})$ as

$$\mathbb{Q}_T(x_{0:t} \mid x_{t+1:T}) \approx \sum_{n=1}^N \tilde{w}_t^n \delta_{x_{0:t}^n}(x_{0:t}),$$

where the *smoothing weights* $\tilde{w}_t^n$ are given by

$$\tilde{W}_t^n = \frac{\mathbb{Q}_T(x_{0:t}^n, x_{t+1:T})}{\mathbb{Q}_t(x_{0:t}^n)} \, w_t^n, \quad \tilde{w}_t^n = \tilde{W}_t^n / \sum_{n=1}^N \tilde{W}_t^n. \tag{11}$$

Thus, we need to compute the ratios $\mathbb{Q}_T(x_{0:t}^n, x_{t+1:T})/\mathbb{Q}_t(x_{0:t}^n)$ in addition to the filtering weights $w_t^n$. The full backward sampling procedure is as follows:

1. Sample a particle at the final time step: $S_T \sim \text{Categorical}(N, \{w_T^n\}_{n=1}^N)$ and $x_T = x_T^{S_T}$.

2. Then for $t \in \{T-1, \ldots, 0\}$, sample an ancestor $S_t \sim \text{Categorical}(N, \{\tilde{w}_t^n\}_{n=1}^N)$ and set $x_{t:T} = (x_t^{S_t}, x_{t+1:T})$, with the smoothing weights $\tilde{w}_t^n$ computed using (11).

## C.2 Details for the Nested SMC Algorithm

We now discuss how to compute the smoothing weights for our nested SMC algorithm (Algorithm 2). Observe that one "particle" of the outer particle filter is the set $\{z_t, a_{t-1}, B_{t-1}^{1:M}, s_t^{1:M}\} =: \Theta_t$, defined for every $t \in \{1, \ldots, T\}$. When doing backward sampling, at time $t$, we will have already sampled a trajectory $\Theta_{t+1:T}$, and we need to sample an ancestor $\Theta_{0:t}^n$ for some $n \in \{1, \ldots, N\}$. Denoting the distribution of $\Theta_{0:t}^n$ by $\Psi_t^M$, the fraction we need to sample an ancestor for time $t$ is given by

$$\frac{\Psi_T^M(\Theta_{0:t}^n, \Theta_{t+1:T})}{\Psi_t^M(\Theta_{0:t}^n)} = \frac{\Psi_T^M(z_{0:t}^n, a_{0:t-1}^n, B_{0:t-1}^{n1:M}, s_{0:t}^{n1:M}, z_{t+1:T}, a_{t:T-1}, B_{t:T-1}^{1:M}, s_{t+1:T}^{1:M})}{\Psi_t^M(z_{0:t}^n, a_{0:t-1}^n, B_{0:t-1}^{n1:M}, s_{0:t}^{n1:M})}$$

$$\propto \underbrace{\prod_{s=t}^{T-1} \pi_\phi(a_s \mid z_{0:t}^n, a_{0:t-1}^n, z_{t+1:s}, a_{t:s-1})}_{\text{Probability of sampling future actions}} \tag{12}$$

$$\prod_{m=1}^M \underbrace{w_t^{nB_t^m}}_{\text{Probability of sampling } B_t^m} \overbrace{f(s_{t+1}^m \mid s_t^{nB_t^m}, a_t)}^{\text{Transition probability for the state}} \quad .$$

Note that we have only written down the terms that are unique to each ancestor $\Theta_{0:t}^n$, as the terms that are common for all $n$ will be normalized away. By using (12) and the stored filtering weights, we can now perform backward sampling for Algorithm 2 as described in Appendix C.1.

The algorithm developed thus far, while correct, can be improved in two major ways. First, when we compute the probability of transitioning from a belief state $\{s_t^{nm}, w_t^{nm}\}_{m=1}^M$ to a belief state $\{s_{t+1}^m, 1/M\}_{m=1}^M$, the expression in (12) looks at the pairwise alignment of individual particles (i.e., the probability of sampling $s_{t+1}^m$ from $s_t^{nB_t^m}$). This is likely to yield very low probabilities for all $n$ except the previously traced ancestor index $A_{t-1}$, because even if the particle sets represent similar posterior distributions, their pairwise alignment can be arbitrarily poor. To fix this, we use a solution proposed in Iqbal et al. [2024] and integrate over the the resampling indices $B_t^{1:M}$, yielding the following transition probability for belief states

$$p(s_{t+1}^{1:M} \mid s_t^{n1:M}, a_t) = \int \prod_{m=1}^M f(s_{t+1}^m \mid s_t^{nB_t^m}, a_t) \, \mathrm{d}\Psi_t^M(B_t^{1:M})$$

$$= \prod_{m=1}^M \int f(s_{t+1}^m \mid s_t^{nB_t^m}, a_t) \, \mathrm{d}\Psi_t^M(B_t^{1:M})$$

$$= \prod_{m=1}^M \left\{ \sum_{k=1}^M w_t^{nk} f(s_{t+1}^m \mid s_t^{nk}, a_t) \right\}. \tag{13}$$

In the second line, we have used the fact that the resampling indices are sampled independently from a categorical distribution. We replace the second line in (12) with (13).

The second improvement we make to the algorithm concerns its computational complexity. Computing the smoothing weights with (12) and 13 for a single $n$ at a single time step $t$ has complexity $\mathcal{O}(M^2 T)$, leading to the full backward sampling algorithm having complexity $\mathcal{O}(N M^2 T^2)$ to simulate a single trajectory. We improve this to $\mathcal{O}(M^2 T^2)$ by using a modified version of backward sampling from Bunch and Godsill [2013], in which the ratio in (12) needs to be computed for only two, rather than $N$, possible ancestors. For details, we refer to Dau and Chopin [2023].

## D    Evaluation Details

### D.1    Architectures

For history-dependent policies, used in both P3O and SLAC, we use a gated recurrent unit [GRU, Cho et al., 2014], a type of recurrent neural network [RNN, Elman, 1990], to encode the history into a fixed-size vector (Table 1), similar to Yang and Nguyen [2021]. This embedding is passed to a multi-layer perceptron (MLP) to get the mean $m$ and standard deviation $\sigma$ of the action distribution (Table 2). Actions are then sampled from a diagonal Gaussian distribution, $a_t \sim \mathcal{N}(m, \mathrm{diag}(\sigma^2))$.

Table 1: The GRU encoder architecture used for P3O and SLAC.

| Layer | Size | Activation |
|---|---|---|
| Input | $\dim(\mathcal{Z} \times \mathcal{A})$ | - |
| Dense | 256 | ReLU |
| LayerNorm | - | - |
| Dense | 256 | ReLU |
| LayerNorm | - | - |
| Dense | 128 | - |
| LayerNorm | - | - |
| GRU | 128 | - |
| GRU | 128 | - |
| Dense | 128 | - |

Table 2: The MLP decoder architecture used for the policies of all algorithms except P3O. For P3O, the noise is independent of the input, and hence the output dense layer has size $\dim(\mathcal{A})$.

| Layer | Size | Activation |
|---|---|---|
| Input | Variable | - |
| Dense | 256 | ReLU |
| Dense | 256 | ReLU |
| Dense | $2 \times \dim(\mathcal{A})$ | - |

The policy in DVRL uses the belief state, which is a weighted particle set, as input. We concatenate the particles and their weights, flatten them, and use a dense layer to encode the belief state as a vector of size 32. This encoded vector is then passed to the decoder in Table 2 to generate the mean and standard deviation of the Gaussian action distribution. The process for DualSMC is similar, except we do not include the weights because the particles are resampled before being passed to the policy, and we append the mean of the particles to the belief state before flattening [Wang et al., 2020].

For the belief-dependent policy for P3O, we use the *set transformer* architecture from Lee et al. [2019] to encode the belief state. The set transformer ensures that the network output is invariant to permutations of the particles in the belief state. The encoded belief state is passed to the same MLP decoder in Table 2.

Finally, SLAC, DualSMC, and DVRL also have critic networks. The critic networks for SLAC and DualSMC have the same architecture, given in Table 3. The critic networks for DVRL use the belief state as input (encoded in the same way as for the policy), and the architecture is given in Table 4.

Table 3: Critic network architecture for SLAC and DualSMC. The input is state, action, and time. The input state is randomly sampled from the belief distribution.

| Layer | Size | Activation |
|-------|------|------------|
| Input | $\dim(\mathcal{S} \times \mathcal{A}) + 1$ | - |
| Dense | 256 | ReLU |
| Dense | 256 | ReLU |
| Dense | 1 | - |

Table 4: Critic network architecture for DVRL. The input is the encoded belief state, action, and time.

| Layer | Size | Activation |
|-------|------|------------|
| Input | $32 + \dim(\mathcal{A}) + 1$ | - |
| Dense | 256 | ReLU |
| Dense | 256 | ReLU |
| Dense | 1 | - |

## D.2 Training Details

SLAC, DualSMC, and DVRL all use soft actor-critic [SAC, Haarnoja et al., 2018] as the base RL algorithm. Our SAC implementation and the hyperparameters chosen for training are based on the implementations in CleanRL [Huang et al., 2022] and Brax [Freeman et al., 2021]. Please see the code for full hyperparameter specification for all reference algorithms.

For all algorithms considered (including P3O), we use a particle filter with 32 particles to track the belief state. Additionally, for P3O, the outer particle filter has 128 particles. Unlike in Algorithm 1, which would use all 128 trajectories to perform one gradient step, we use mini-batches to perform multiple gradient updates per nested SMC run to improve sample efficiency.

For P3O, we use an additional slew rate penalty in the reward function that penalizes large changes in action in adjacent time steps. We found that this was necessary to ensure that the trajectories sampled using Algorithm 2 are smooth enough for the GRU networks to learn.

All our experiments were carried out on an NVIDIA A100 80GB GPU. For complete details, including the scripts used for the experiments, see the source code at `https://github.com/Sahel13/particle-pomdp`.

