# OpenReview forum: "Sequential Monte Carlo for Policy Optimization in Continuous POMDPs"
_NeurIPS.cc/2025/Conference — NeurIPS 2025 poster_

### Official Review · Reviewer_6N23 · 2025-06-30

**Clarity:** 3
**Significance:** 3
**Originality:** 3
**Rating:** 5
**Confidence:** 3

**Summary:**

The paper addresses the problem of solving planning problems in POMDPs. For this ideas from particle filtering and control via inference are combined. The authors present a nested particle filtering approach to solve the planning problem under partial observability. The paper evaluates the problem on a set of benchmarks problems.

**Questions:**

How could you deal with unknown models in this work, such as, reward functions, dynamics or emission probabilities?

**Ethical Concerns:**

["NO or VERY MINOR ethics concerns only"]

**Final Justification:**

I have read the rebuttal and thank the authors for their comments. I appreciate the inclusion of the LQG benchmark and further discussion on the algorithms details. I will keep my score and recommendation.

**Limitations:**

yes

**Paper Formatting Concerns:**

-

**Quality:**

3

**Strengths And Weaknesses:**

The paper is overall well-written and presents an interesting method that demonstrates its performance on benchmark tasks. The related work is integrated nicely and provides a solid foundation for the reader. The explanation of related concepts is thorough, although in some cases—such as the detailed description of the bootstrap filter or control via inference—it may be more elaborate than necessary for the main narrative.

The method itself is compelling and shows promise, particularly in its ability to handle complex scenarios. However, several areas could benefit from further development. For instance, the derivation of the core methodology is largely relegated to the appendix; incorporating more of this into the main text would enhance clarity and accessibility. Additionally, a deeper discussion of computational complexity is warranted, especially given that the nesting approach implies an exponential increase in the number of particles required.

Empirical results are promising, but the evaluation could be strengthened. More extensive results for the filter would be valuable, particularly to demonstrate the absence of variance collapse. Including reference results for standard models such as LQG would also help contextualize performance. Furthermore, it would be beneficial to show experimentally how the proposed estimator achieves lower variance, as claimed.

The manuscript raises important questions that are not fully addressed, such as how the method scales to high-dimensional settings, what specific requirements are imposed by the approach, and how it might be adapted to handle unknown models—potentially through learning-based techniques like the ideas presented within DVRL.

Despite these points, the paper makes a meaningful contribution to the field and introduces a novel approach that is both technically sound and practically relevant. The strengths of the work outweigh the current limitations, and I believe it will spark valuable discussion and further research.

---

> ### Author Rebuttal · Authors · 2025-07-30
>
> We thank the reviewer for their time and evaluation. We provide responses to the concerns raised below.
>
> > A deeper discussion of computational complexity is warranted, especially given that the nesting approach implies an exponential increase in the number of particles required.
>
> The reviewer is correct to highlight the computational complexity issue. We would like to clarify, the nesting does not necessarily imply exponential growth in particle number, instead it leads to complexity $N \times M$ per time step. Nonetheless, based on the comments of multiple reviewers, we believe the issue of complexity is important and has to be addressed within the main text. We have added the following text to the discussion in Section 6:
>
> *Algorithm 2 has computational complexity $\mathcal{O}(NMT)$, which increases to $\mathcal{O}(NM^2T^2)$ when employing backward sampling (Appendix C). For long-horizon problems, we therefore recommend avoiding backward sampling and instead tuning the parameter $\eta$ to mitigate particle degeneracy while maintaining computational efficiency.*
>
> Additionally, in our answer to Reviewer [hAe2], we discuss further details on how the number of particles scales with dimensionality in SMC methods in general.
>
> ---
>
> > Including reference results for standard models such as LQG would also help contextualize performance. Furthermore, it would be beneficial to show experimentally how the proposed estimator achieves lower variance, as claimed.
>
> Based on the reviewer's recommendation, we have now included an LQG baseline comparison. The baseline uses LQG to optimally control a linear-Gaussian dynamical system with quadratic rewards accompanied by a Kalman filter belief-tracker. The resulting plot shows the optimal performance of LQG and how our policy learning method gradually converges to the same level over learning iterations, thus empirically validating against the closed-form LQG baseline.
>
> Additionally, we run an experiment to analyze the variance of the gradient estimate for the LQG problem. The table below clearly shows that lower values of $\eta$ lead to higher variance of the policy gradient as measured by the trace of the gradient covariance matrix. It is important to remark that the REINFORCE gradient is recovered at the limit of $\eta \to 0$, corresponding to high-variance gradients:
>
> | Tempering $\eta$   | Trace |
> |----------------------|----------------------------------------|
> | $ 1.0 \times 10^{-3} $ | $ 21.2627 \pm 0.6912 $ |
> | $ 5.0 \times 10^{-3} $ | $ 20.0137 \pm 0.7931 $ |
> | $ 1.0 \times 10^{-2} $ | $ 19.7993 \pm 0.6831 $ |
> | $ 5.0 \times 10^{-2} $ | $ 15.5139 \pm 0.3298 $ |
> | $ 1.0 \times 10^{-1} $ | $ 13.3801 \pm 0.3931 $ |
> | $ 5.0 \times 10^{-1} $ | $ 6.3718 \pm 0.2818 $ |
> | $ 1.0 \times 10^{0} $  | $ 4.2700 \pm 0.1835 $ |
> | $ 5.0 \times 10^{0} $  | $ 1.7832 \pm 0.0647 $ |
> | $ 1.0 \times 10^{1} $  | $ 1.2546 \pm 0.0505 $ |
>
> Finally, we run more ablation studies to highlight the role of the number of particles $N$ and $M$ on the learning procedure. The plots show an expected behavior: increasing $N$, $M$, leads to better approximations of the distribution $\Psi_{T}$ and contributes to faster learning and better quality solutions.
>
> **Unfortunately, the NeurIPS rebuttal guidelines do not allow us to provide a modified PDF with the highlighted changes.**
>
> ---
>
> >How could you deal with unknown models in this work, such as, reward functions, dynamics or emission probabilities?
>
> This is an interesting question. On the one hand, our approach does not require full access to reward and dynamics functions. These can be treated as black-box quantities to sample transitions and receive reward signals. However, our method currently requires oracle access to the observation model in order to evaluate emission probabilities while tracking the belief.
>
> In principle, it is feasible to postulate an extension of our method that learns model, similar to [1, 2]. One can parameterize a conditional density estimator with a variational auto-encoder or a normalizing flow and alternate between policy optimization and density-learning in Algorithm 2. This may require formulating and optimizing a lower-bound on the marginal log-likelihood, but the details of such an approach are difficult to explore in the scope of this rebuttal.
>
> We appreciate the reviewer recognizing our contribution nonetheless. We deliberately focused on the case with known models to isolate and address the fundamental challenge of POMDP optimization without resorting to the suboptimal QMDP approximation typically used in RL frameworks. Our future work will target making this approach as general as possible.
>
> ---
>
> ## References:
>
> 1. Lee, A. X., Nagabandi, A., Abbeel, P., & Levine, S. (2020). Stochastic latent actor-critic: Deep reinforcement learning with a latent variable model. NeurIPS.
> 2. M. Igl, L. Zintgraf, T. A. Le, F. Wood, and S. Whiteson (2018). Deep variational reinforcement learning for POMDPs. ICML.

---

> > ### Comment · Reviewer_6N23 · 2025-08-06
> >
> > I have read the rebuttal and thank the authors for their comments. I appreciate the inclusion of the LQG benchmark and further discussion on the algorithms details. I will keep my score and recommendation.

---

> ### Author Response · Authors · 2025-08-07
>
> We thank the reviewer for recognizing our contribution. Their comments on complexity, LQG, and empirical variance evaluation were very helpful. Those changes will appear in the final draft.

---

### Official Review · Reviewer_uahs · 2025-07-03

**Clarity:** 3
**Significance:** 3
**Originality:** 2
**Rating:** 4
**Confidence:** 3

**Summary:**

This paper presents a nested particle filtering framework for policy optimization in partially observable Markov decision processes (POMDPs). The authors formulate a risk-sensitive belief-space objective using a Feynman–Kac distribution and derive a novel policy gradient estimator (Proposition 2) based on Fisher’s identity. The proposed method performs sampling over belief trajectories using a two-layer particle filter—one for observation-action histories and another for latent state inference. The framework is validated on continuous-control and visual POMDP benchmarks.

**Questions:**

1. Discuss the bias–variance tradeoff in the nested sampling procedure. How does the number of history particles (N) and belief particles (M) affect performance and stability?
2. Provide empirical or theoretical insights into whether the gradient estimator is prone to local minima due to limited expressiveness or the nested approximation.
3. I am not an expert in the policy learning area. Please clarify whether this method supports online or only offline training, and what implications this approach may have for real-time control.

**Ethical Concerns:**

["NO or VERY MINOR ethics concerns only"]

**Limitations:**

Yes

**Quality:**

3

**Strengths And Weaknesses:**

Strengths

1. Clear and Coherent Framework: The paper presents a well-motivated and mathematically coherent approach to optimizing policies in POMDPs via nested sequential Monte Carlo (SMC). The derivation of the Feynman–Kac-based trajectory distribution is sound.
2. Main Innovation in Proposition 2: The derivation of the policy gradient using Fisher’s identity over the Feynman–Kac belief-trajectory distribution appears novel in the context of POMDPs. This formulation offers a clean way to estimate gradients under a risk-sensitive trajectory objective.
3. Use of Risk-Sensitive Objective: The incorporation of a soft-optimality reward structure (Equation 6) is relatively uncommon in belief-space POMDP literature and can be viewed as a meaningful design contribution.
4. Algorithm Design: The nested structure in Algorithm 2 is a practical realization of the theoretical framework, clearly leveraging a two-level particle system to handle both history uncertainty and belief uncertainty.

Weaknesses
1. Limited Novelty Outside Proposition 2: While Proposition 2 is the core contribution, most other components rely on known techniques. For example, Proposition 1 simply reformulates the standard POMDP objective in belief space, and the use of Feynman–Kac models in SMC is well-established.
2. Lack of Discussion on Optimization Challenges: The method relies on gradient ascent over a non-convex and highly stochastic objective. The paper does not discuss whether the optimization suffers from local minima or high variance in gradient estimates, especially under particle approximation and finite horizon.
3. Scalability and Cost: Although the algorithm is elegant, the nested particle structure introduces significant computational cost (O(NM) per time step). There is no discussion on scaling the method to long-horizon or high-dimensional settings.
4. Missing Comparisons: While comparisons to some baselines are provided, the paper omits relevant prior works that also perform sampling-based policy learning in belief or latent space (e.g., VIREL, Variational Policy Search, SMCP, PO-SMC).
5. No Ablation or Variance Analysis: It would be valuable to show the effect of η (risk level), particle count (N, M), or nesting on gradient variance and convergence.

---

> ### Author Rebuttal · Authors · 2025-07-30
>
> We thank the reviewer for their time and evaluation. We provide responses to the concerns raised below.
>
> >Scalability and Cost: Although the algorithm is elegant, the nested particle structure introduces significant computational cost (O(NM) per time step). There is no discussion on scaling the method to long-horizon or high-dimensional settings.
>
> The reviewer raises a valid point. We previously discussed time complexity in the appendix in Section C.2. However, given its importance, we agree it should be highlighted in Section 6. We have now added the following text to Section 6:
>
> *Algorithm 2 has computational complexity $\mathcal{O}(NMT)$, which increases to $\mathcal{O}(NM^2T^2)$ when employing backward sampling (Appendix C). For long-horizon problems, we therefore recommend avoiding backward sampling and instead tuning the parameter $\eta$ to mitigate particle degeneracy while maintaining computational efficiency.*
>
> ---
>
> We also agree with the reviewer that scalability to high dimensions is an issue. Our draft briefly raises this point in Section 6. However, given the reviewer's comment, we have decided to expand that discussion to highlight the point further. Section 6 now includes the following text:
>
> *The bootstrap particle filter used in our implementation may not scale well in high-dimensional settings due to increased discrepancy between the proposal and target distributions. Using more sophisticated proposals that take the likelihood into account [3, Chapter 10] or learning better proposals [4, 5] may be necessary to ensure satisfactory performance for high-dimensional problems.*
>
> Please also see our answer to Reviewer [hAe2] for a more detailed discussion on scaling to higher dimensions.
>
> ---
>
> > While comparisons to some baselines are provided, the paper omits relevant prior works that also perform sampling-based policy learning in belief or latent space (e.g., VIREL, Variational Policy Search, SMCP, PO-SMC).
>
> We thank the reviewer for pointing out potentially relevant prior work that we may have overlooked. To our knowledge, VIREL [1] and Variational Policy Search [2] are algorithms that target MDP settings and are not applicable to POMDPs. As for SMCP and PO-SMC, despite our efforts, we were unable to identify papers with these specific acronyms in the POMDP literature. We would be grateful if the reviewer could provide full citations or author names, as these works may use different terminology than we anticipated.
>
> ---
>
> > No Ablation or Variance Analysis: It would be valuable to show the effect of $\eta$ (risk level), particle count (N, M), or nesting on gradient variance and convergence.
>
> Following the reviewer's recommendation, we have now conducted ablation studies examining the impact of key hyperparameters on algorithm performance. Specifically, we performed ablation analyses in the Cart-Pole and Light-Dark experiments, varying the following hyperparameters:
>
> | **Cart-Pole** | **Light-Dark** |
> |---------------|----------------|
> | Outer particles: $N \in$ \{32, 64, 128\}  | Outer particles: $N \in$ \{32, 128, 256\} |
> | Inner particles: $M \in$ \{32, 128, 512\}  | Inner particles: $M \in$ \{16, 64, 256\} |
> | Tempering: $ \eta \in$ \{0.005, 0.01, 0.05, 1\} | Tempering: $ \eta \in$ \{0.001, 0.005, 0.01, 0.05\} |
>
> In this ablation study, we plot the learning progress for the listed hyperparameters, averaged over 10 seeds. The plots show an expected behavior: increasing $N$, $M$, leads to better approximations of the distribution $\Psi_{T}$ and contributes to faster training. The parameter $\eta$ controls the variance of the gradient: lower values of $\eta$ lead to higher-variance gradients and cause learning to stall and increase the variability across seeds.
>
> To isolate the role of $\eta$ even further, we run an additional experiment to analyze the variance of the gradient estimate in a linear-quadratic-Gaussian (LQG) control problem. The table attached below clearly shows that lower values of $\eta$ lead to higher variance of the policy gradient as measured by the trace of the gradient covariance matrix. It is important to remark that the REINFORCE policy gradient is recovered for $\eta \to 0$, corresponding to high-variance policy gradients:
>
> | Tempering $\eta$   | Trace |
> |----------------------|----------------------------------------|
> | $ 1.0 \times 10^{-3} $ | $ 21.2627 \pm 0.6912 $ |
> | $ 5.0 \times 10^{-3} $ | $ 20.0137 \pm 0.7931 $ |
> | $ 1.0 \times 10^{-2} $ | $ 19.7993 \pm 0.6831 $ |
> | $ 5.0 \times 10^{-2} $ | $ 15.5139 \pm 0.3298 $ |
> | $ 1.0 \times 10^{-1} $ | $ 13.3801 \pm 0.3931 $ |
> | $ 5.0 \times 10^{-1} $ | $ 6.3718 \pm 0.2818 $ |
> | $ 1.0 \times 10^{0} $  | $ 4.2700 \pm 0.1835 $ |
> | $ 5.0 \times 10^{0} $  | $ 1.7832 \pm 0.0647 $ |
> | $ 1.0 \times 10^{1} $  | $ 1.2546 \pm 0.0505 $ |
>
> These ablation studies provide important practical guidance for hyperparameter selection. The detailed results and figures have been added to Section 5 of the revised manuscript. We are grateful to the reviewer for this suggestion, which has improved the quality and completeness of our experimental results.
>
> **Unfortunately, the NeurIPS rebuttal guidelines do not allow us to provide a modified PDF with the highlighted changes.**
>
> ---
>
> >Discuss the bias–variance tradeoff in the nested sampling procedure. How does the number of history particles (N) and belief particles (M) affect performance and stability?
>
> In standard particle filters, both bias and variance decrease at $\mathcal{O}(1/N)$ rate with the number of particles $N$ [3], so there is no traditional bias-variance tradeoff with respect to the particle count. We expect the same property to hold for our nested particle filter: increasing either $N$ or $M$ reduces both bias and variance.
>
> The primary bias-variance tradeoff in our method arises from the temperature parameter $\eta$, where lower values provide unbiased but high-variance gradient estimates, while higher values reduce variance at the cost of introducing bias with respect to the risk-neutral objective. This behavior is standard in decision-making objectives that incorporate risk.
>
> For a more detailed comment of how $N$ and $M$ affect performance and stability, please see our response to your comment on ablation analysis above.
>
> ---
>
> > Provide empirical or theoretical insights into whether the gradient estimator is prone to local minima due to limited expressiveness or the nested approximation.
>
> We assume the reviewer meant "limited expressiveness *of* the nested approximation".
> Our nested SMC algorithm introduces bias in our gradient estimates, but it is difficult to draw conclusions on the quality of the solutions based on that. Biases appear in most policy optimization techniques, for example, in bootstrapped TD learning that powers most state-of-the-art algorithms, or in approaches that rely on iterative linearizations, or variational Gaussian approximations, which are popular approaches in POMDPs. Like most policy gradient methods, our approach can converge to local optima, mainly because the underlying decision-making problem is non-convex, but this non-convexity is inherent to the general POMDP structure. There is no indication that the nested SMC structure exacerbates this issue compared to the other POMDP solution methods we compare to.
>
> ---
> >Please clarify whether this method supports online or only offline training, and what implications this approach may have for real-time control.
>
> Our approach involves directly interacting with the environment during training so it is online in that sense. We generate new trajectories using the current policy, perform updates and iterate over this procedure. However, once a policy is learned, deployment is very cheap. Obtaining actions only requires a forward pass through a neural network, leading to minimal computational requirements, and making our approach practical for real-time control applications.
>
> ---
> ## References:
>
> 1. Fellows, M., Mahajan, A., Rudner, T. G., Whiteson, S. (2019). VIREL: A variational inference framework for reinforcement learning. NeurIPS.
>
> 2. Levine, S., Koltun, V. (2013). Variational policy search via trajectory optimization. NeurIPS.
>
> 3. Chopin, N., \& Papaspiliopoulos, O. (2020). An introduction to sequential Monte Carlo. Springer.
>
> 4. Naesseth, C., Linderman, S., Ranganath, R., \& Blei, D. (2018). Variational sequential Monte Carlo. AISTATS.
>
> 5. Heng, J., Bishop, A. N., Deligiannidis, G., \& Doucet, A. (2020). Controlled sequential Monte Carlo. The Annals of Statistics, 48(5), 2904-2929.

---

> > ### Comment · Reviewer_uahs · 2025-08-03
> >
> > I maintain my evaluation and rating for this work. Overall, it is a solid paper, and I would be happy to see it accepted. I would welcome more demonstrations of its practical applications to POMDP problems.

---

> ### Author Response · Authors · 2025-08-07
>
> We appreciate the positive evaluation. The reviewer's comments contributed to clarifying important points in the experiments and discussion sections. The promised changes will appear in the final draft.

---

### Official Review · Reviewer_hAe2 · 2025-07-03

**Clarity:** 3
**Significance:** 2
**Originality:** 3
**Rating:** 4
**Confidence:** 3

**Summary:**

This paper proposes a new policy optimization framework for continuous POMDPs, named P3O. It addresses a key problem where many existing reinforcement learning methods, relying on the QMDP approximation, cannot perform deliberate information gathering because they ignore the value of future observations. To overcome this, the study formulates the POMDP as a probabilistic inference problem within a Feynman-Kac mode. This approach inherently incorporates the value of information gathering by anticipating future observations. The framework uses a nested Sequential Monte Carlo (SMC) method for optimization, which efficiently estimates the policy gradient by sampling from optimal trajectories. Experiments confirm that P3O significantly outperforms existing methods, particularly on tasks that require deliberate information gathering.

**Questions:**

**My primary concern is P3O's dependence on a true observation model**, which significantly limits its applicability to real-world problems. Could the authors elaborate on the feasibility of experimentally demonstrating an extension where this dependency is removed by using a learned, rather than an oracle, observation model?

**Ethical Concerns:**

["NO or VERY MINOR ethics concerns only"]

**Final Justification:**

The authors' rebuttal successfully addressed my concerns regarding scalability. The proposed method is interesting, but its fundamental dependence on a true (oracle) observation model remains a significant limitation for practical, real-world applications. While the work has merit, this key constraint makes it a borderline case.

**Limitations:**

yes

**Paper Formatting Concerns:**

I found no formatting concerns

**Quality:**

3

**Strengths And Weaknesses:**

## Strengths

- The paper presents a coherent theoretical framework by reformulating policy learning in POMDPs as a  probabilistic inference problem within a non-Markovian Feynman-Kac model, based on the connection between optimization and inference.
- The proposed method, P3O, avoids the QMDP approximation and reasons directly in the belief space, enabling it to incorporate actions aimed at reducing future uncertainty into its plans. This strength is clearly demonstrated in tasks where information gathering is key to success, such as 'Light-Dark' and Triangulation. The experiments show that agents trained with P3O learn to perform intentional exploration—such as first moving toward a light source ("Light-Dark") or executing an S-shaped maneuver ("Triangulation") to reduce positional uncertainty, rather than moving directly to the goal. This provides strong evidence for P3O's superiority, as these are tasks where QMDP-based methods fail.

## Weaknesses

- **Concerns about Scalability to High-Dimensional Problems.** As the authors acknowledge, the core of the proposed P3O method is a nested Sequential Monte Carlo (SMC) algorithm. This approach is susceptible to the "curse of dimensionality," where the number of particles required increases exponentially as the dimensionality of the state and observation spaces grows, leading to a significant degradation in performance.
- **Strong Dependence on a True Environment Model (Oracle).** P3O's learning loop uses a particle filter to track the belief state. This filter assumes that the true observation model, g(z|s), which maps states to observations, is known and accessible (i.e., as an oracle). This assumption is not met in many realistic scenarios, such as robotics or finance, where the environment model is often unknown. Therefore, the applicability of the method in its current form is significantly limited. The paper lacks discussion and experiments on how the method's performance would change if the observation model itself were learned from data.

---

> ### Author Rebuttal · Authors · 2025-07-30
>
> We thank the reviewer for their evaluation. We provide responses to the concerns raised below.
>
> > Concerns about Scalability to High-Dimensional Problems. As the authors acknowledge, the core of the proposed P3O method is a nested Sequential Monte Carlo (SMC) algorithm. This approach is susceptible to the "curse of dimensionality," where the number of particles required increases exponentially as the dimensionality of the state and observation spaces grows, leading to a significant degradation in performance.
>
> The reviewer raises an important point about scalability. They are correct that SMC methods, including our nested SMC approach, can suffer from the curse of dimensionality in high-dimensional problems due to its reliance on importance sampling. We appreciate the opportunity to clarify our perspective on this important limitation.
>
> In many practical problems with high-dimensional state/observation spaces, the likelihood is concentrated in lower-dimensional manifolds, and the effective dimension governing particle requirements can be much smaller than the ambient dimension [1]. The number of particles required is also influenced by the discrepancy between the target and proposal distributions [1, 2], and hence can be reduced by either constructing or learning better proposals that are adapted to the target [3, 4, 5]. Algorithm 2 admits the use of any proposal distribution, and our choice of the transition dynamics as the proposal was made to to simplify the exposition. Future extensions of this work can focus on integrating proposal learning into the policy learning loop.
>
> We have modified the discussion section in the paper as follows to incorporate these ideas:
>
> *The bootstrap particle filter used in our implementation may not scale well in high-dimensional settings due to increased discrepancy between the proposal and target distributions. Using more sophisticated proposals that take the likelihood into account [3, Chapter 10] or learning better proposals [4, 5] may be necessary to ensure satisfactory performance for high-dimensional problems.*
>
> ---
>
> > My primary concern is P3O's dependence on a true observation model, which significantly limits its applicability to real-world problems. Could the authors elaborate on the feasibility of experimentally demonstrating an extension where this dependency is removed by using a learned, rather than an oracle, observation model?
>
> As we acknowledge in the draft, we agree with the reviewer that the dependence on a true observation model limits our algorithm's applicability. However, we would like to highlight that there are many real-world POMDP settings with known observation models. For example, in robotic systems, like manipulators and drones, only position measurements are available while velocity cannot be measured. Another example is bearings‑only active target tracking in radar systems. In both cases, observation models can be extracted analytically from physical relationships.
>
> Nonetheless, in principle, it is feasible to postulate an extension of our method that learns an observation model, similar to [6, 7]. One can parameterize a conditional density estimator with a variational auto-encoder or a normalizing flow and alternate between policy optimization and observation-density learning in Algorithm 2. Unfortunately, the exact details cannot be reasonably explored in the scope of this rebuttal.
>
> We deliberately focused on the case with known models to isolate and address the fundamental challenge of POMDP optimization without resorting to the suboptimal QMDP approximation typically used in RL frameworks.
>
> ---
>
> ## References
>
> 1. Agapiou, S., Papaspiliopoulos, O., Sanz-Alonso, D., \& Stuart, A. M. (2017). Importance sampling: Intrinsic dimension and computational cost. Statistical Science, 405-431.
> 2. Chatterjee, S., \& Diaconis, P. (2018). The sample size required in importance sampling. The Annals of Applied Probability, 28(2), 1099-1135.
> 3. Chopin, N., \& Papaspiliopoulos, O. (2020). An introduction to sequential Monte Carlo. Springer.
> 4. Naesseth, C., Linderman, S., Ranganath, R., \& Blei, D. (2018). Variational sequential Monte Carlo. AISTATS.
> 5. Heng, J., Bishop, A. N., Deligiannidis, G., \& Doucet, A. (2020). Controlled sequential Monte Carlo. The Annals of Statistics, 48(5), 2904-2929.
> 6. Lee, A. X., Nagabandi, A., Abbeel, P., & Levine, S. (2020). Stochastic latent actor-critic:
> Deep reinforcement learning with a latent variable model. NeurIPS.
> 7. M. Igl, L. Zintgraf, T. A. Le, F. Wood, and S. Whiteson (2018). Deep variational reinforcement learning for
> POMDPs. ICML.

---

> ### Comment · Reviewer_hAe2 · 2025-08-06
>
> Thank you for your thoughtful rebuttal, and please accept my apologies for the delayed response.
>
> The clarifications have addressed many of my concerns, and I found the rebuttal largely convincing. While I understand and appreciate the authors' perspective on the need for a true observation model, this remains my primary concern, as I believe it narrows the scope of the method's application in real-world scenarios.
>
> Nevertheless, the rebuttal has strengthened the paper, and I will increase my score to reflect this (3->4).

---

> > ### Author Response · Authors · 2025-08-07
> >
> > We sincerely thank the reviewer for giving our rebuttal a serious consideration. The points they raised will be highlighted in the discussion section in the final draft.

---

### Official Review · Reviewer_CaZk · 2025-07-03

**Clarity:** 2
**Significance:** 3
**Originality:** 4
**Rating:** 5
**Confidence:** 3

**Summary:**

The paper presents novel techniques for Reinforcement Learning (RL) in *continuous* Partially-Observable Markov Decision Processes (POMDPs)—one of the most central and challenging problems in this area.
First, the paper proposes an innovative technique based on results that have not been so far applied to POMDPs. Specifically, it reframes the RL problem for POMDPs in terms of Feynman–Kac Models, which also provides an alternative conceptual perspective on the problem
Then, based on the formulation above, the paper proposes a Sequential Monte Carlo (SMC) algorithm, which allows for providing discrete approximations to Feynman–Kac models.
Finally, the above contributions provide the basis for a *novel policy gradient method for POMDPs*.
The method is evaluated empirically on four standard benchmarks, comparing it against the state-of-the-art methods SLAC, DVRL, and Dual-SMC, *showing significant improvements*.

**Questions:**

**1.** Do you think finite bounds on performance guarantees (e.g., PAC bounds or regret bounds), or at least convergence guarantees, can be proved?

**2.** If convergence cannot be guaranteed in the general case, do you see any reasonable assumptions that might suffice to ensure convergence? Or in general, conditions under which the problem might be well-behaved from a theoretical and/or practical point of view?

**3.** Do you think your technique could be relevant for Regular Decision Processes (RDPs)* with continuous observations and actions?
*Reference on RDPs: *Offline RL in Regular Decision Processes: Sample Efficiency via Language Metrics.* Deb et al., ICLR 2025.

**Ethical Concerns:**

["NO or VERY MINOR ethics concerns only"]

**Final Justification:**

The authors have addressed all my concerns during the rebuttal, and promised to integrate the changes they have described. I will keep my positive score.

**Limitations:**

No limitations beyond my comments above.

**Paper Formatting Concerns:**

- Equation (1): Replace equality with definition symbol ':=', as the notation $\mathcal{B}(\phi)$ has not yet been defined. An alternative is to introduce it earlier in a definition environment.

**Quality:**

3

**Strengths And Weaknesses:**

## Strengths

- The paper addresses a **very central problem**. It is in fact one of the most central problems in the POMDP literature. It is a challenging problem, for which no 'ultimate' solution is so far available, and improvements on the state-of-the-art are always important and likely to be impactful, both research-wise and in applications.

- The **technical development is rigorous**.

- The proposed **techniques are innovative** in that they combine several techniques that were not considered before in addressing RL in POMDPs, e.g., non-Markovian Feynman–Kac models.

- The proposes **techniques are empirically shown to be effective**, improving on the state-of-the-art in challenging problems. Notably, a large improvement is observed in an active triangulation task.


## Weaknesses


**No performance guarantees.** The technique is not accompanied by a theoretical analysis deriving performance guarantees (e.g., PAC bounds or regret bounds) or at least convergence guarantees.

**Presentation style can be improved.** The presentation style focuses on pointing out how exsiting frameworks and results are combined together in order to develop the novel contribution, with few words spent on describing the resulting novel definitions, and constructions.  As an example, in Line 143 and Equation (4), the intuitive meaning and the role of the auxiliary random variables $\mathcal{O}_t$ is not given. Also, the choice of the exponential distribution in Equation (4) is not discussed.  With this presentation style, I feel that all the reader can do is to mechanically follow the paper as pieces come together, and wait to get to the end of the paper (making a significant effort) to build intuitions and the bigger pictures on how all pieces come together, and why specific choices were made in the constructions. Even worse, the presentation style may make the paper accessible only to an audience that is already familiar with all the employed notions and frameworks. This is far from ideal, as I understand that the paper is rather inter-disciplinary in its nature, as its novelty is largely based on addressing a central (classic) problem of POMDPs by resorting to techniques that have not been used before in this field.

---

> ### Author Rebuttal · Authors · 2025-07-30
>
> We thank the reviewer for their positive assessment and valuable suggestions for strengthening the manuscript. We address each comment and question in detail below.
>
> > The presentation style focuses on pointing out how existing frameworks and results are combined together in order to develop the novel contribution, with few words spent on describing the resulting novel definitions, and constructions. As an example, in Line 143 and Equation (4), the intuitive meaning and the role of the auxiliary random variables is not given. Also, the choice of the exponential distribution in Equation (4) is not discussed.
>
> We thank the reviewer for this feedback on improving the clarity of our presentation. Defining an auxiliary random variable with a Boltzmann/Gibbs distribution is a common trick in reinforcement learning and the control-as-inference literature [1, 2, 3], but we agree that the current presentation assumes too much familiarity with this approach. We have added the following sentences to the text to address your concerns:
>
>   *For time $t$, $\\{ \\mathcal{O}\_{t} = 1 \\}$ represents the event that the pair $(z_{t}, a_{t-1})$ is optimal given the history $(z_{0:t-1}, a_{0:t-2})$. A higher probability of optimality corresponds to higher expected reward $\ell_{t}$. The exponential function is strictly positive and monotonic. Positive potential functions are required by the Feynman-Kac formalism and monotonicity guarantees that the potential function maintains the same maximizer as the reward function.*
>
> We also acknowledge the reviewer's broader point about the presentation style making readers work through all details before seeing the complete picture. We will make small edits throughout the manuscript to improve flow and clarity where possible, while maintaining the technical structure.
>
> ---
>
> > Do you think finite bounds on performance guarantees (e.g., PAC bounds or regret bounds), or at least convergence guarantees, can be proved? If convergence cannot be guaranteed in the general case, do you see any reasonable assumptions that might suffice to ensure convergence? Or in general, conditions under which the problem might be well-behaved from a theoretical and/or practical point of view?
>
> Providing convergence guarantees for our method is challenging. Our algorithm employs stochastic gradient ascent to maximize the objective $\\mathcal{B}_{\\eta}(\\phi)$. One of the two requirements for convergence to a local optimum under the Robbins-Monro conditions is an unbiased gradient estimator. However, our gradient estimates are computed as expectations over a particle approximation obtained with an SMC algorithm, which is biased for any finite number of particles [4].
>
> While standard SMC methods have well-understood bias properties, $\\mathcal{O}(1/N)$ with $N$ particles for additive functionals [4, 5], our nested structure with $N$ outer particles and $M$ inner particles creates more complex bias behavior whose theoretical characterization is still an open problem. Despite this theoretical limitation, the algorithm demonstrates reliable behavior in practice. We observe that increasing both $N$ and $M$ improves performance, and we can control gradient variance through the temperature parameter $\eta$. To support these claims empirically, we have added ablation studies that highlight the role of $N$ and $M$ on the learning progress, and on the role of $\eta$ in decreasing the gradient variance. Please see our answer to Reviewer [uahs] for more details on these empirical results.
>
> Formal PAC or regret bounds are beyond the scope of this work. Future theoretical contributions could include quantifying the bias of our nested SMC algorithm as a function of $N$ and $M$ (similar to [6]). Understanding this bias-particle relationship could potentially lead to convergence guarantees under appropriate growth conditions on $N$ and $M$ as learning progresses.
>
> ---
>
> > Do you think your technique could be relevant for Regular Decision Processes (RDPs)* with continuous observations and actions? *Reference on RDPs: Offline RL in Regular Decision Processes: Sample Efficiency via Language Metrics. Deb et al., ICLR 2025.
>
> We thank the reviewer for bringing RDPs to our attention, which we were not aware of. Given our limited experience with the RDP literature [7, 8], we want to be cautious with our claims. However, upon initial review of the RDP framework, it seems that RDPs can be understood as POMDPs [8] with finite observation and action spaces, and dynamics representable by finite-state automata. While we designed our algorithm for continuous POMDPs where the states, observations and actions lie in continuous spaces, we think the approach is general and can accommodate discrete or discrete-continuous POMDPs, and may be applicable to RDPs with continuous observations and actions.
>
> ---
>
> ## References
> 1. Levine, S. (2018). Reinforcement learning and control as probabilistic inference: Tutorial and review. arXiv preprint arXiv:1805.00909.
> 2. Lee, A. X., Nagabandi, A., Abbeel, P., \& Levine, S. (2020). Stochastic latent actor-critic: Deep reinforcement learning with a latent variable model. NeurIPS.
> 3.  Lázaro-Gredilla, M., Ku, L., Murphy, K. P., \& George, D. (2024). What type of inference is planning?. NeurIPS.
> 4. Del Moral, P. (2004). Feynman-Kac formulae: Genealogical and interacting particle systems with applications. Springer New York.
> 5. Kantas, N., Doucet, A., Singh, S. S., Maciejowski, J., \& Chopin, N. (2015). On particle methods for parameter estimation in state-space models. Statistical Science, 30(3).
> 6. Crisan, D., \& Miguez, J. (2018). Nested particle filters for online parameter estimation in discrete-time state-space Markov models. Bernoulli, 24(4a).
> 7. Deb, A., Cipollone, R., Jonsson, A., Ronca, A., \& Talebi, M. S. (2025). Offline RL in Regular Decision Processes: Sample Efficiency via Language Metrics. ICLR
> 8. Brafman, R. I., \& De Giacomo, G. (2019). Regular Decision Processes: A Model for Non-Markovian Domains. IJCAI.

---

> > ### Comment · Reviewer_CaZk · 2025-08-06
> >
> > I would like to thank the reviewers for the detailed and informative response to my concerns and questions. The response fully addressed all my points. I encourage the authors to improve the manuscript by integrating their responses to my point. I will keep my positive score.

---

> ### Author Response · Authors · 2025-08-07
>
> We appreciate the reviewer taking the time to engage with the the paper and rebuttal. The promised changes based on their comments will appear in the final draft.

---

### Decision · Program_Chairs · 2025-09-17

**Decision:**

Accept (poster)

**Comment:**

This paper introduces a novel formulation of RL in continuous POMDPs via Feynman–Kac models, leading to a Sequential Monte Carlo algorithm and a new policy gradient method. The approach is original and shows empirical gains over QMDP baselines.

Reviewers agree that importing Feynman–Kac models into RL is both interesting and relevant. Continuous POMDPs are a challenging problem, and this new perspective is promising.

Concerns were raised about scalability to high-dimensional problems, reliance on a true observation model, and the thoroughness of the experimental evaluation. The rebuttal provided additional results that alleviated some of the experimental concerns. Theoretical and scalability aspects were addressed, but there is still clear scope for further work.

Thus, I recommend acceptance, while encouraging the authors to strengthen both theoretical guarantees and practical scalability to realistic high-dimensional problems moving forward.